# Evolution of longitudinal division in multi-cellular bacteria of the *Neisseriaceae* family

Sammy Nyongesa[1,9], Philipp M. Weber [2,3,9], Ève Bernet [1], Francisco Pulido[1], Cecilia Nieves [1], Marta Nieckarz [4], Marie Delaby[5], Tobias Viehboeck [2,3,6], Nicole Krause [2,3], Alex Rivera-Millot[1], Arnaldo Nakamura [1], Norbert O. E. Vischer[7], Michael vanNieuwenhze [8], Yves V. Brun [5], Felipe Cava[4], Silvia Bulgheresi [2,10] ✉ & Frédéric J. Veyrier [1,10] ✉

Rod-shaped bacteria typically elongate and divide by transverse fission. However, several bacterial species can form rod-shaped cells that divide longitudinally. Here, we study the evolution of cell shape and division mode within the family *Neisseriaceae*, which includes Gram-negative coccoid and rod-shaped species. In particular, bacteria of the genera *Alysiella*, *Simonsiella* and *Conchiformibius*, which can be found in the oral cavity of mammals, are multicellular and divide longitudinally. We use comparative genomics and ultrastructural microscopy to infer that longitudinal division within *Neisseriaceae* evolved from a rod-shaped ancestor. In multicellular longitudinally-dividing species, neighbouring cells within multicellular filaments are attached by their lateral peptidoglycan. In these bacteria, peptidoglycan insertion does not appear concentric, i.e. from the cell periphery to its centre, but as a medial sheet guillotining each cell. Finally, we identify genes and alleles associated with multicellularity and longitudinal division, including the acquisition of amidase-encoding gene *amiC2*, and amino acid changes in proteins including MreB and FtsA. Introduction of *amiC2* and allelic substitution of *mreB* in a rod-shaped species that divides by transverse fission results in shorter cells with longer septa. Our work sheds light on the evolution of multicellularity and longitudinal division in bacteria, and suggests that members of the *Neisseriaceae* family may be good models to study these processes due to their morphological plasticity and genetic tractability.

[1]INRS-Centre Armand-Frappier Santé Biotechnologie, Bacterial Symbionts Evolution, Laval, QC H7V 1B7, Canada. [2]Department of Functional and Evolutionary Ecology, Environmental Cell Biology Group, University of Vienna, Vienna, Djerassiplatz 1, 1030 Vienna, Austria. [3]University of Vienna, Vienna Doctoral School of Ecology and Evolution, Vienna, Austria. [4]Department of Molecular Biology and Laboratory for Molecular Infection Medicine Sweden (MIMS), Umeå Centre for Microbial Research (UCMR), Umeå University, Umeå SE-90187, Sweden. [5]Département de microbiologie, infectiologie et immunologie, Université de Montréal, Montréal, QC, Canada. [6]Division of Microbial Ecology, Center for Microbiology and Environmental Systems Science, , University of Vienna, Djerassiplatz 1, 1030 Vienna, Austria. [7]Bacterial Cell Biology & Physiology, Swammerdam Institute of Life Sciences, Faculty of Science, University of Amsterdam, Science Park 904, 1098 Amsterdam, the Netherlands. [8]Indiana University, Bloomington, IN 47405, USA. [9]These authors contributed equally: Sammy Nyongesa, Philipp M. Weber. [10]These authors contributed equally: Silvia Bulgheresi, Frédéric J. Veyrier. ✉e-mail: silvia.bulgheresi@univie.ac.at; frederic.veyrier@inrs.ca

Allometry of animal-microbe associations suggests that $10^{25}$ prokaryotes thrive on animals and $10^{23}$ on humans[1,2]. Yet, the morphology and growth mode of animal symbionts are underexplored[3]. Although many may form biofilms (see for example refs. 4, 5), intestinal segmented filamentous bacteria (SFB[6-8]) and three genera of *Neisseriaceae* that occur in the oral cavity (e.g., *Alysiella*, *Simonsiella* and *Conchiformibius*[9-12]), are the only known animal symbionts that may be regarded as multicellular, i.e., they invariably form stable filaments of more than two cells. SFB occur in the small intestine of several animals and play a primal role in pathogen resistance and gut homeostasis[13,14]. In contrast to SFB, multicellular oral cavity *Neisseriaceae* are relatively understudied. They are closely related to the other ≈30 species of *Neisseriaceae* occurring, for the majority, in the buccal cavity of warm-blooded vertebrates. They are cultivable and some are genetically tractable[15,16]. Apart from being multicellular, *Neisseriaceae* may be rod-shaped (e.g., *Neisseria elongata*) or coccoid (e.g., the human pathogens *Neisseria meningitidis* and *Neisseria gonorrhoeae*). *Alysiella filiformis* cells are 2 μm-long and 0.6 μm-wide on average and form upright-standing palisades on the squamous epithelium of the mouth, so that each cell has a proximal pole attached to the host epithelium and a distal, free pole (Figs. 1, 2b, and Supplementary Fig. 2a–d). Furthermore, within each filament, *A. filiformis* cells appear as paired (Fig. 2b, Supplementary Figs. 1c and 2a). Concerning *Simonsiella muelleri* and *Conchiformibius steedae* (previously known as *Simonsiella*

steedae[11]) they are thinner, but can be up to 4 and 7 μm-long, respectively. Unlike *A. filiformis*, both poles of *S. muelleri* and *C. steedae* are attached to the mouth[11,17]. This confers *S. muelleri* and *C. steedae* cells a curved (or crescent-shaped) morphology and we will henceforth refer to their host-attached poles as proximal and to their midcell as their most-distal region (Figs. 1, 2c, d; Supplementary Figs. 1d, e and 2d–f).

Besides multicellularity, another peculiarity of *Alysiella*, *Simonsiella* and *Conchiformibius* is that they divide longitudinally ([11,18,19] and this manuscript). This is extraordinary, given that, except for nematode[20,21], insect[22] and dolphin symbionts[23], rod-shaped bacteria typically elongate and divide by transverse fission, two processes coordinated by the elongasome and divisome, respectively. In model bacteria, each of these machineries is constituted by over a dozen proteins, with the actin homolog MreB and the tubulin homolog FtsZ, respectively, orchestrating cell elongation and division[24]: deletion of *ftsZ* results in filamentation[25], whereas inactivation of *mreB* turned rods into cocci[16,26]. Even more striking was the effect of specific amino acid changes: in MreB, they resulted in irregularly sized, bent or branched *Escherichia coli* cells[27,28] and, when affecting FtsZ, they led to misplaced septa in *E. coli*, *Bacillus subtilis* and *Streptomyces* spp.[29-31]. Curiously, single amino acid mutations in the FtsZ-binding protein SsgB resulted in longitudinally dividing *Streptomyces*[32]. Collectively, these findings led to the hypothesis that longitudinal division might have evolved from differential regulation of subtly different MreB and/or FtsZ variants[33,34].

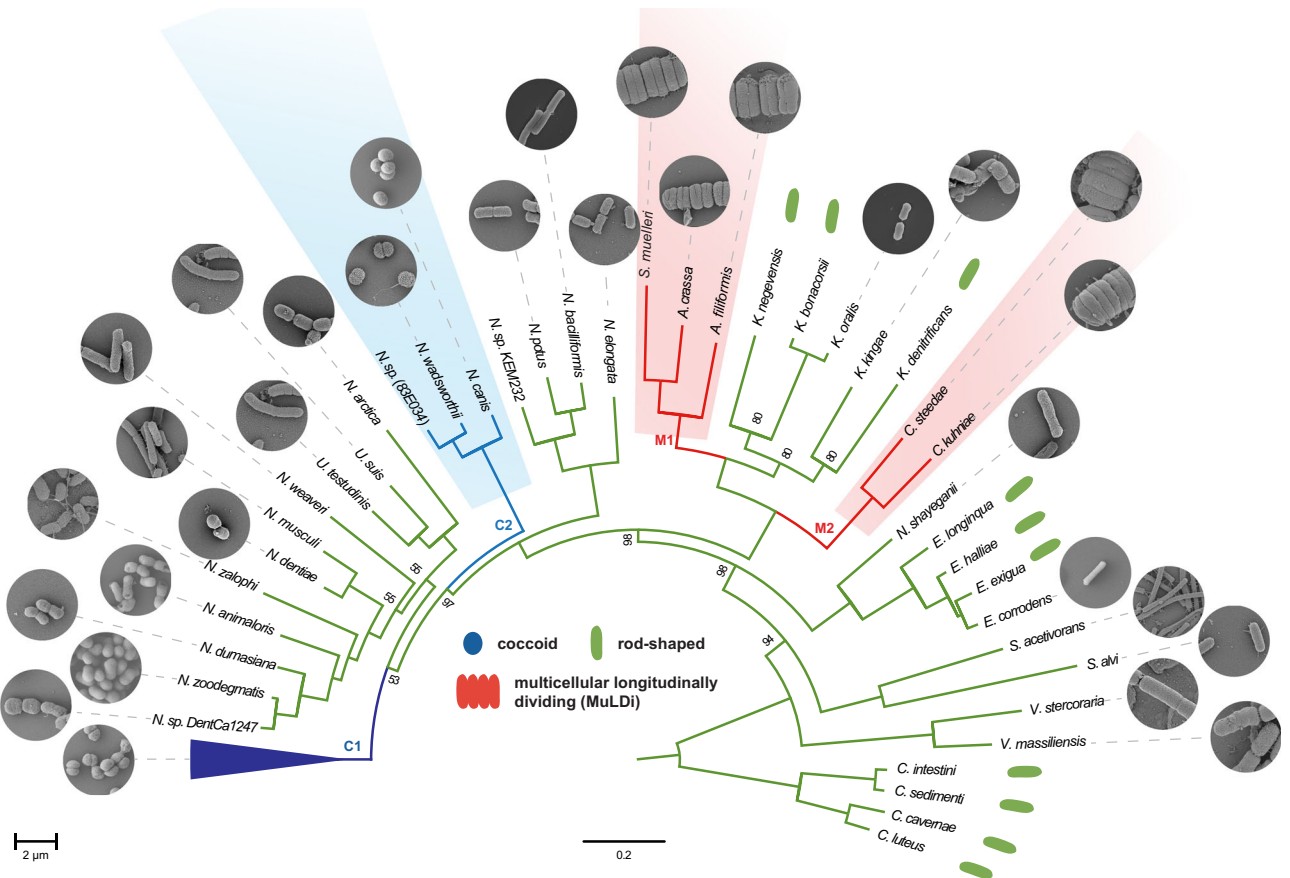

**Fig. 1 | Core genome-based phylogeny of rod-shaped, coccoid and MuLDi *Neisseriaceae*.** The best evolutionary model for each partition was found by IQ-TREE version 1.6.3[81] and maximum-likelihood phylogenetic analysis was also performed using IQ-TREE[82] using 10,000 ultrafast bootstrap replicates[83]. Above the name of each species, scanning electron microscopy images display their morphology. Dark and light blue: coccoid *Neisseriaceae*; green: rod-shaped *Neisseriaceae*; red: multicellular longitudinally dividing (MuLDi) *Neisseriaceae*. Coccoid lineages 1 and 2 are indicated in blue. MuLDi lineages 1 and 2 are indicated in red. N.: *Neisseria*; U.: *Uruburuella*; S.: *Simonsiella*; A.: *Alysiella*; K.: *Kingella*; C.: *Conchiformibius*; S.: *Snodgrassella*; V.: *Vitreoscilla*; E.: *Eikenella*; C.: *Crenobacter*. *Crenobacter* spp. served as outgroup. In the absence of electron microscopy images, species' morphology was defined as rod-shaped based on the reference strain describe in refs. 94 for *K. negevensis*[95], for *K. bonacorsii*[96], for *K. denitrificans*[97], for *E. longiqua* and *E. haliae*[98], for *Crenobacter luteus*[99], for *C. cavernae*[100], for *C. sedimenti*[101], for *C. intestine*.

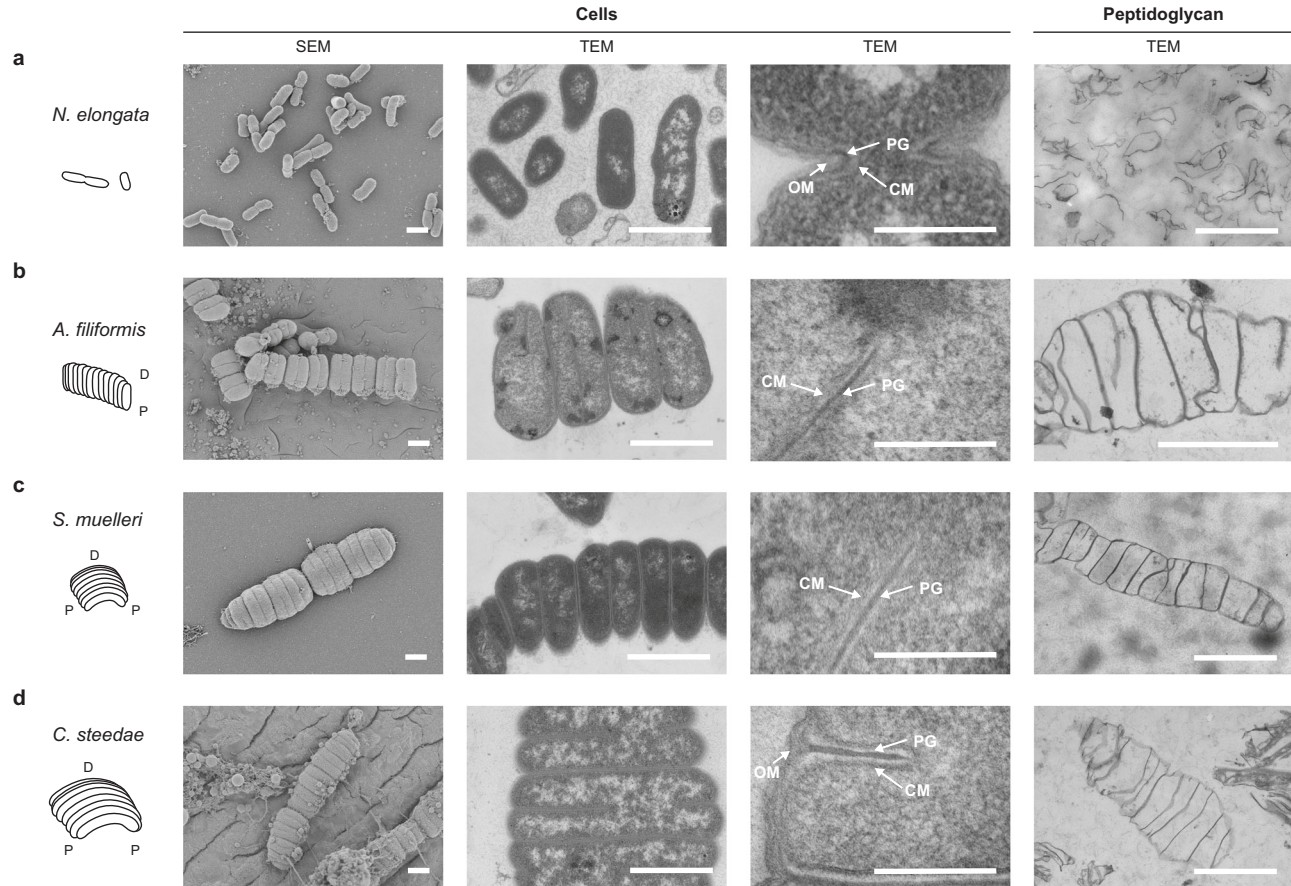

**Fig. 2 | Ultrastructural analysis of four oral cavity symbionts belonging to the *Neisseriaceae*.** Schematic representations and electron microscope images of **a** *N. elongata*, **b** *A. filiformis*, **c** *S. muelleri* and **d** *C. steedae*. Rightmost panels display extracted sacculi (peptidoglycan) of respective *Neisseriaceae*. P proximal (host-attached) region of the cell, D distal region of the cell, OM outer membrane, PG peptidoglycan, CM cytoplasmic membrane. Scale bars correspond to 1 μm. The results are representative of at least three independent analyses.

In this work, we ask whether a similar path led to the evolution of *Alysiella*, *Simonsiella* and *Conchiformibius* – henceforth, collectively referred to as multicellular longitudinally dividing (MuLDi) *Neisseriaceae*. Phylogenomic analysis coupled with both ultrastructural analysis and peptidoglycan (PG) mass spectrometry indicates that MuLDi *Neisseriaceae* evolved from a rod-shaped ancestor. Moreover, incubation with a palette of fluorescent D-amino acids (FDAAs) shows that nascent septa cross the cells medially as to guillotine them – from the proximal to the distal pole in *A. filiformis*, or from both poles to midcell in *S. muelleri* and *C. steedae*. Finally, comparative genomics-informed recapitulation of MuLDi-specific allelic changes in the rod-shaped *N. elongata* results in longer septa, suggesting that the transition from transverse to longitudinal division required the deletion of *mraZ*, the acquisition of *amiC2* and MreB amino acid permutations.

The capacity of oral cavity *Neisseriaceae* to have evolved – more than once – into coccoid or MuLDi cells from a rod-shaped ancestor, together with their amenability to cultivation and genetic manipulation, makes them ideal models to understand the evolution of bacterial cell division and that of animal-bacterium symbioses.

## Results

### Core genome-based phylogeny of *Neisseriaceae* suggests that MuLDi *Neisseriaceae* evolved from a rod-shaped ancestor

The *Neisseriales* order comprises the family *Chromobacteriaceae* and the family *Neisseriaceae* and more recently three additional families have been suggested, *Aquaspirillaceae*, *Chitinibacteraceae* and *Leeiaceae*[35]. The family *Neisseriaceae* includes 12 genera (*Alysiella*; *Bergeriella*; *Conchiformibius*; *Eikenella*; *Kingella*; *Morococcus*; *Neisseria*;

*Simonsiella*; *Snodgrassella*; *Stenoxybacter*; *Uruburuella*; *Vitreoscilla*). We selected species from each of these *Neisseriaceae* genera and used SMRT (PacBio) and Minion (Nanopore) technologies to obtain 21 closed genomes (Supplementary Data 1). Genomes obtained in this study were combined with *Neisseriaceae* draft genomes (a total of 262, only one strain of *N. meningitidis* and one of *N. gonorrhoeae* were included) from the NCBI database to calculate the Average Nucleotide Identity (ANI) (Supplementary Data 2). This enabled us to identify 75 *Neisseriaceae* species with genome ANI > 96%. To assure the quality of the genomic database and to simplify the genomic comparisons, we then selected one genome for each species based on (1) completeness and circularization status, and (2) possibility to morphologically characterize it by either using a strain we have in our collection or by literature search (in the case of morphologically characterized reference strains). These 75 genomes have been used for the construction of a core genome-based phylogeny (Fig. 1, Supplementary Fig. 1, Supplementary Data 3). Of note, although most of the genomes available in the NCBI database are from coccoid *Neisseria* (lineage 1; dark blue in Fig. 1, representing 34 species), the detailed phylogenetic analysis of this lineage, which evolved from an ancestral rod[16] and includes the well-known pathogens *N. meningitidis* and *N. gonorrhoeae*, will be presented elsewhere (Bernet and Veyrier, unpublished data). In this manuscript, we therefore focused on the analysis of the remaining 41 species (Fig. 1, Supplementary Data 1).

Using Scanning-Electron Microscopy (SEM), we imaged all the species available in public collections to classify them as rod, cocci or MuLDi. The cell-shape of 10 species was already known (see references in Fig. 1's legend). Of note, species that could not be unambiguously

classified as rods or cocci, were incubated in sublethal concentrations of Penicillin G to test their elongation capacity, as previously described[16]. These morphological analyses revealed that most *Neisseriaceae* are rod-shaped, except for the previously identified coccoid lineage 1[16] and the two closely related species *N. wadsworthii* and *N. canis*. These did not lengthen upon Penicillin G treatment and are henceforth referred to as coccoid lineage C2 (light blue branches in Fig. 1). Remarkably, we found that coccoid species belonging to lineage 2 harbour genes encoding for the elongasome, but lost *yacF/zapD* (Fig. 5). The loss of *yacF/zapD* has already been described as a major genetic event, which also allowed the emergence of coccoid lineage 1[16]. We also observed that MuLDi species are separated into two lineages, henceforth referred to as MuLDi lineage M1 (*Alysiella* spp. and *Simonsiella muelleri*) and M2 (*Conchiformibius* spp.) with the monophyletic *Kingella* genus separating them, in agreement with a recently published study[35]. To extrapolate the shape of the ancestor of all *Neisseriaceae*, we used a Maximum Likelihood method (PastML[36]) which made us infer that the predecessor of all *Neisseriaceae* was a rod (see Supplementary Fig. 1). This conclusion is supported by the fact that species belonging to the closely related family *Chromobacteriaceae* (order *Neisseriales*, as aforementioned) are also described as rod-shaped[37].

Collectively, our phylogenetic analysis indicates that two lineages of cocci (coccoid lineages 1 and 2, referred to as C1 and C2 in Fig. 1) evolved independently from a rod-shaped ancestor, whereas the two lineages of MuLDi *Neisseriaceae* (referred to as M1 and M2 in Fig. 1) evolved from a rod-shaped ancestor. However, PastML-based analysis (Supplementary Fig. 1) was not able to determine the shape of the most recent common ancestor of M1, M2 and *Kingella* spp. This let us envision two evolutionary scenarios: (1) M1 and M2 evolved independently from a rod-shaped bacterium with phenotypic convergence or (2) the common ancestor of M1 (*Simonsiella/Alysiella*), *Kingella* spp. and M2 (*Conchiformibius*) evolved the MuLDi phenotype once from a rod-shaped bacterium, but *Kingella* spp. reverted to unicellularity and transverse division.

## MuLDi *Neisseriaceae* cells are attached to one another by their lateral PG and harbour a characteristic signature in their muropeptide composition

Previous[11,18,19], as well as our, microscopic analyses (see Figs. 1–4, Supplementary Figs. 2c–e, 3–8, and Supplementary Movies 1–4) suggested that *A. filiformis*, *S. muelleri* and *C. steedae* filaments result from incomplete cell separation. Moreover, Nile red staining confirmed the presence of membranes between adjoining cells (Supplementary Fig. 4a, d). To understand whether adjoining MuLDi *Neisseriaceae* share additional cellular structures, that prevent them to separate from one another, we performed transmission electron microscopy (TEM) of sacculi extracted from *A. filiformis*, *S. muelleri* and C. *steedae*, as well as from the transversally dividing rod-shaped *N. elongata*, for comparison (rightmost panels in Fig. 2b–d). We observed that the sacculi of the three MuLDi symbionts remained attached laterally, even after the harsh extraction procedure (rightmost panels in Fig. 2b–d). Moreover, higher magnification of TEM images revealed that cells belonging to the same filament shared their outer membrane (OM; arrows in Fig. 2b–d). We concluded that in the *Neisseriaceae A. filiformis*, *S. muelleri* and *C. steedae* multicellularity results from adjoining cells, retaining their cytoplasmic membranes (CM), but being attached to one another by their lateral PG and surrounded by a common OM (and periplasm) (see Supplementary Fig. 3).

We previously showed that a modification in the PG composition of the *Neisseriaceae* (i.e., increased proportion of pentapeptides) accompanied their rod-to-coccus transition[16]. To find out whether the rod-to-MuLDi transition would also correlate with a change in total muropeptide composition, we applied mass spectrometry to analyze the PG of three MuLDi: *A. filiformis*, *S. muelleri* and C. *steedae*, as well as

that of 14 rod-shaped *Neisseriaceae* (Supplementary Fig. 5, Supplementary Data 4). The abundance of dimers (Di), trimers (Tri) and tetramers (Tetra) relative to the abundance of monomers and the estimated total crosslinked were generally higher in MuLDi (Supplementary Fig. 5c, d). We concluded that, compared to rod-shaped *Neisseriaceae*, MuLDi *Neisseriaceae* PG was more cross-linked (Supplementary Fig. 5).

## *Alysiella filiformis* nascent septa guillotine the cells from their distal to their proximal poles

Fimbriae-like structures were detected by TEM on the regions of *A. filiformis* attached to oral epithelial cells[17–19]. To confirm the presence of fimbriae at the proximal pole, we immunostained them with an anti-fimbriae antibody and found its signal to be localized at the proximal pole, consistent with the seminal ultrastructural data. Moreover, we noticed that, when observed at the epifluorescence microscope, the proximal, fimbriae-rich side of each filament was invariably the convex one (Supplementary Fig. 4a–c, f), which allowed us to determine *A. filiformis* polarity in the absence of fimbriae localization in all the subsequent microscopic analyses.

After confirming *A. filiformis* polarity, we proceeded to determine its growth mode by tracking PG synthesis by consecutively applying the three fluorescent D-amino acids (FDAAs) HADA (blue), BADA (green) and TADA (red), which are labeled D-Ala residues incorporated into the peptide side chains of new PG. When observed by fluorescence microscopy, *A. filiformis* cells sequentially labeled with HADA 30 min, BADA 15 min and TADA 15 min showed strongest fluorescent signal at their septation planes. The virtual time-lapse obtained by the triple FDAAs labeling revealed that *A. filiformis* starts to septate at the distal pole and that PG synthesis proceeded unidirectionally toward the proximal cell pole (blue, green and red signal in representative septa in Fig. 3a). Of note, within each filament, newly completed septa (asterisks) alternate with nascent septa (arrowheads in Fig. 3a right panel). This indicates that septation starts as soon as cells are born (or even before). Plotting the total fluorescence of HADA, BADA and TADA against the cell long axis in a representative nascent septum (septum 1; Fig. 3b, left plot), in an almost completed septum (septum 2; Fig. 3b right plot), as well as in ten *A. filiformis* cells undergoing ten subsequent septation stages (Supplementary Fig. 6a–c) confirmed the distal-to-proximal PG incorporation pattern (Fig. 3c) and was consistent with that observed by thin-section TEM (Fig. 2b).

To view the PG insertion pattern in 3D, we performed confocal microscopy (Fig. 3d, Supplementary Fig. 6d and Supplementary Movie 5). Surprisingly, the septal signal appeared as a sheet when viewed from the side in completed and nascent septa (asterisks and arrowheads, respectively, in Fig. 3d left panel; Supplementary Fig. 6d) and, contrarily to what observed by 3D-Structured Illumination Microscopy in other longitudinally[38] or transversally dividing bacteria[39,40], we did not observe PG disks, rings or arcs at any septation stage.

In conclusion, we showed that *A. filiformis* septation is unidirectional (i.e., it proceeds from the distal to the proximal pole) and that the PG is not inserted concentrically, from the periphery to the center of the cell, but as a sheet that guillotined each cell from its distal to its proximal pole.

## *Simonsiella muelleri* and *Conchiformibius steedae* septation starts at both poles synchronously and proceeds from the poles to midcell

Based on previous ultrastructural studies, *S. muelleri* fimbriae are situated on the cell side facing the epithelial cells[17,19], here referred to as the proximal side. To test whether this was also the case for *C. steedae*, we immunostained it with an anti-fimbriae antibody and confirmed that fimbrial appendages covered the proximal (concave) side of each filament (Supplementary Fig. 4e, f).

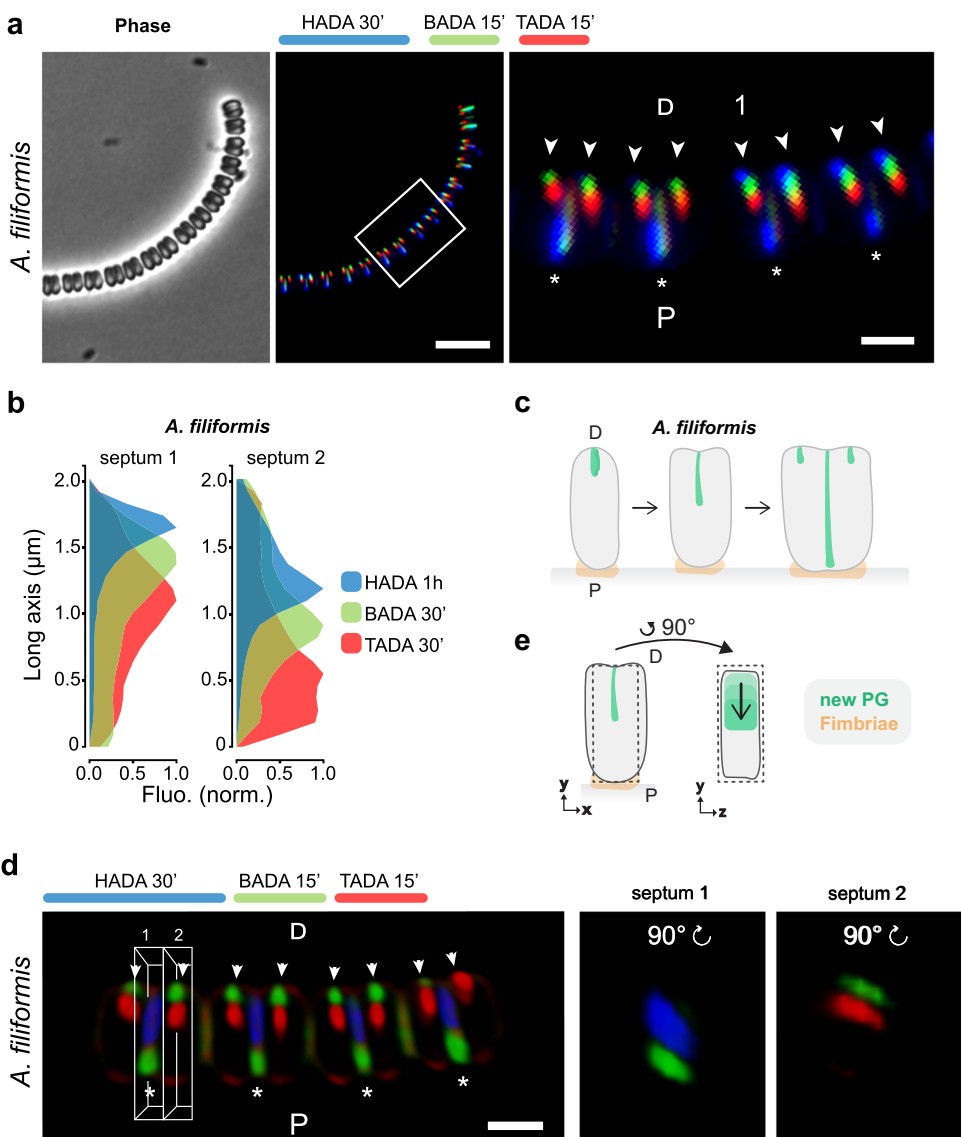

**Fig. 3 | Epifluorescence and confocal microscope-based PG insertion pattern in A. filiformis. a** Phase contrast image (left panel), corresponding epifluorescence image (middle panel) and enlarged selected regions (white frames in right panel) of *A. filiformis* consecutively labeled with HADA, BADA and TADA for 30 min, 15 min and 15 min, respectively. Asterisks point at newly completed septa and arrowheads point to nascent septa. Scale bars are 5 μm (middle panels) and 1 μm (right panels). **b** Septal fluorescence of HADA, BADA and TADA was plotted onto the long axis for two representative *A. filiformis* cells. Source data are provided as a Source Data file. **c** Schematic representation of *A. filiformis* growth mode. **d** Confocal images of *A.* *filiformis* consecutively labeled with HADA, BADA and TADA for 30 min, 15 min and 15 min, respectively. Asterisks point at newly completed septa and arrowheads point at nascent septa in a filament (left panel). Fluorescence emitted by a newly completed septum (septum 1 in white box in left panel) and by an incoming septum (septum 2 in white box in left panel) were rotated by 90° and are displayed in the middle and the right panels, respectively. Scale bar is 1 μm. **e** Schematic representation of *A. filiformis* growth mode. D distal, P proximal. The results are representative of at least three independent analyses.

To find out how *S. muelleri* and *C. steedae* grow, we then tracked the synthesis of PG by successively labeling them in three differently colored FDAAs, namely with HADA for 30 min, BADA 15 min and TADA 15 min, and with HADA for 1 h, BADA 45 min and TADA 45 min, respectively (Fig. 4a–d and Supplementary Fig. 7a–c). When imaged by epifluorescence microscopy, both species showed strongest fluorescent signal at the septation plane. However, the virtual time-lapse obtained by the triple FDAAs labeling differed from that obtained for *A. filiformis*. Namely, *S. muelleri* and *C. steedae*, appeared to start septation at both poles synchronously and PG insertion continued until midcell was reached (Fig. 4a–d and Supplementary Fig. 7a–c).

To view the PG insertion pattern in 3D, we performed confocal microscopy on FDAA- labeled *C. steedae* (Fig. 4e, f and Supplementary Fig. 8a, b; Supplementary Movie 6). When pulsing filaments with HADA, TADA and BADA, septation appeared to begin at the poles and proceeded towards midcell (blue, green and red signal in completed and nascent septa, indicated by asterisks or arrowheads, respectively, in Fig. 4e). When visualizing single cells turned of 90 degrees (Fig. 4e bottom images and Supplementary Fig. 8b), FDAA signal appeared as two juxtaposed triangular sheets, each emerging from one cell pole (green and red signal in septa 1 and 2 Fig. 4e). With septation progression, the two leading edges merged at midcell (red oval signal in Fig. 4e, septum 2) and finally appeared as a circular disk at the very last septation stage (red signal in Fig. 4e, septum 3 and one septum in Supplementary Fig. 6b).

Summarizing, we propose that the two curved oral symbionts *S. muelleri* and *C. steedae* start septation at each pole independently,

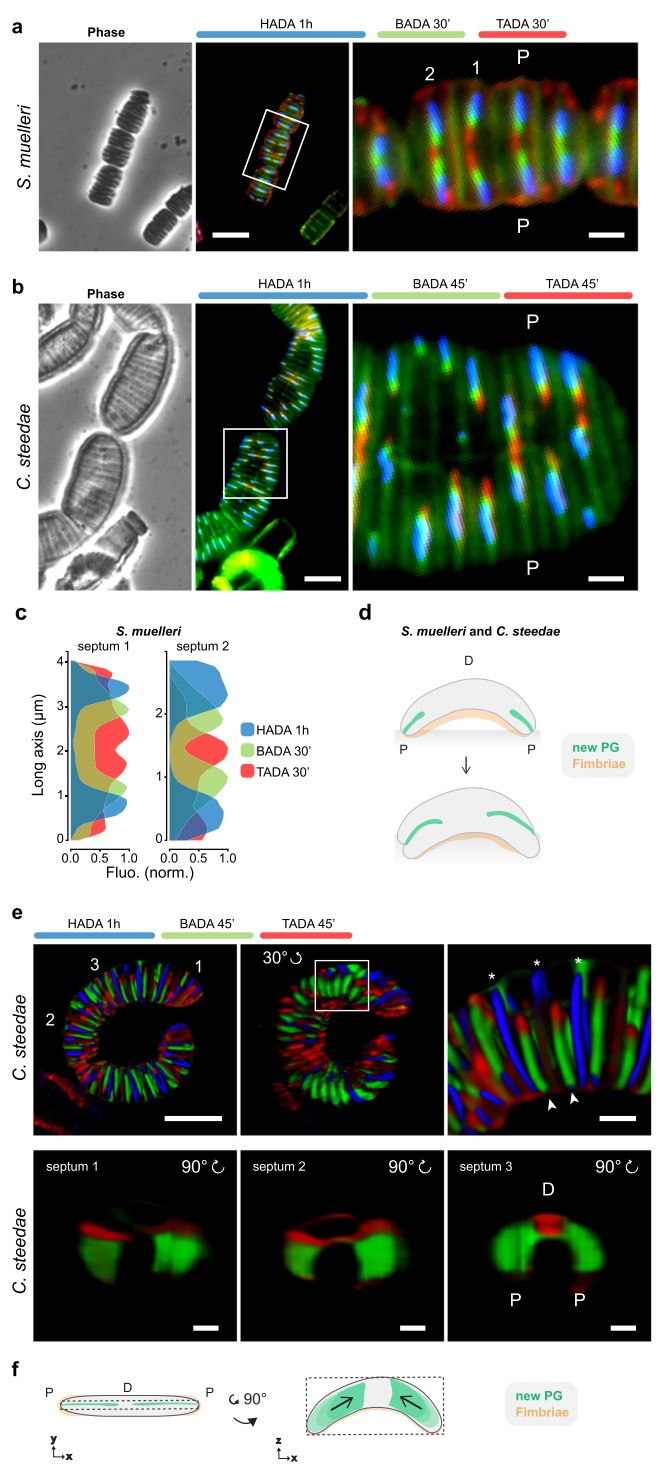

**Fig. 4 | Epifluorescence and confocal microscopy-based PG insertion pattern in *S. muelleri* and *C. steedae*.** Phase contrast images (left panels), corresponding epifluorescence images (middle panels) and enlarged selected regions (right panels; the white frames indicate the selected regions) of *S. muelleri* (**a**) labeled with HADA, BADA and TADA for 1 h, 30 min and 30 min, respectively and of *C. steedae* (**b**) labeled with HADA, BADA and TADA for 1 h, 45 min and 45 min, respectively. Scale bars are 5 μm (middle panels) and 1 μm (right panels). **c** For two representative *S. muelleri* septa (septum 1 and septum 2, left and right panel), fluorescence of HADA, BADA and TADA was plotted onto the long axis. Scale bars are 5 μm (left and middle panel) and 1 μm (right panel). Source data are provided as a Source Data file. **d** Schematic representation of *S. muelleri* and *C. steedae* growth mode. **e** Confocal images of one *C. steedae* filament labeled with HADA, BADA and TADA for 1 h, 45 min and 45 min, respectively (top panels). Top left panel displays the filament from which the three septa shown in the bottom panels belong to. Middle panel shows the same filament rotated by 30° of which an enlarged region of interest (white frame) is shown in the top right panel; arrowheads point to nascent septa, asterisks to newly completed ones. Bottom panels: three septa at consecutive septation stages (septa 1–3 in top left panel) were rotated by 90° and ordered from the youngest to the oldest (left, middle and right panel, respectively). D distal, P proximal. Scale bars are 5 μm (left upper corner) and 1 μm. **f** Schematic representations of *C. steedae* septation mode (top view in left panel, side view in right panel). The results are representative of at least three independent analyses.

but synchronously, and septation ends when the two pole-originated PG sheets meet and merge at midcell (Fig. 4f).

## Multiple genetic events associated with the cell shape transition from rod-shaped to MuLDi *Neisseriaceae*

By applying exhaustive comparative genomics, we previously discovered that mutations at specific genetic loci mediated the rod-to-coccus transition of the ancestor of pathogenic *Neisseria*[16]. We therefore hypothesized that mutations at specific genetic loci had mediated their evolution from an ancestral, transversally dividing rod-shaped *Neisseriaceae* (Fig. 1). To identify these genetic loci, we applied previously described pipelines[16,41,42] to determine the presence/absence of proteins in 37 species of *Neisseriaceae* (all displayed in Fig. 1), 32 rod-

shaped and 5 MuLDi *Neisseriaceae* (the *Simonsiella/Alysiella* lineage M1 and the *Conchiformibius* lineage M2). Of note, we excluded both lineages of coccoid *Neisseriaceae* from our analysis, as they underwent a different evolutionary path[16].

By using MycoHIT (based on tblastn) and by taking either the genome of the MuLDi *S. muelleri* or that of the rod-shaped *N. elongata* as a reference, we searched for genes specifically present in MuLDi or specifically present in rod-shaped *Neisseriaceae*, respectively. Firstly, using *S. muelleri* as a reference and 55% of similarity as a cut-off for assessing orthologs, we identified seven genes that were exclusively present in MuLDi, but absent in rod-shaped *Neisseriaceae* (Fig. 5a). These included a gene encoding for an AmiC-like amidase, henceforth referred to as AmiC2. Interestingly, the *amiC2* gene chromosomally co-locates with *cdsA*, a gene encoding for the phosphatidate cytidylyl-transferase CdsA in all MuLDi species (Supplementary Fig. 9). As *amiC2* and *cdsA* are either flanked by a transposase (in the MuLDi lineage 1) or by a restriction/modification system (in the MuLDi lineage 2), we hypothesize that *amiC2* was acquired by horizontal gene transfer, possibly from a *Fusobacterium*-related bacterium (see AmiC1 and AmiC2 phylogeny in Supplementary Fig. 10). Intriguingly, Fuso-bacteria, as the *Neisseriaceae*, are common members of the oral, gas-trointestinal and genital flora[43]. As for the remaining six MuLDi-specific genes, four are predicted to encode for hypothetical proteins and two for the hemolysin transporter ShlB[44].

Secondly, using *N. elongata* as a reference, we found that only four genes were exclusively absent in MuLDi *Neisseriaceae* when compared to rod-shaped ones (Fig. 5a), two of which, *mraZ* and *rapZ*, have been implicated in PG synthesis and cell division. *mraZ* is the first gene of the *dcw* cluster (Supplementary Fig. 9) in most bacteria, where it encodes for the poorly characterized, but highly conserved transcriptional regulator MraZ[45–47]. *rapZ* encodes for the small RNA adaptor protein RapZ, implicated in cell envelope precursor sensing and signaling[48]. As for the other two, *dgt* and *gloB*, the former encodes for a dGTPase[49] and the latter for a hydroxyacylglutathione hydrolase that hydrolyzes S-D-lactoyl-glutathione into glutathione and D-lactic acid[50]. Of note, as the loss or gain of genes in the most recent common ancestor of the M1, M2 and *Kingella* spp. could have laid the foundation for further evolution, we also detected the presence of 14 genes and the absence of four genes in MuLDi as compared to other rod-shaped *Neisser-iaceae* excluding the *Kingella* spp. (Supplementary Data 5).

Third and finally, we used the CapriB software[41] to search for amino acid changes in the 438 proteins strictly conserved among the

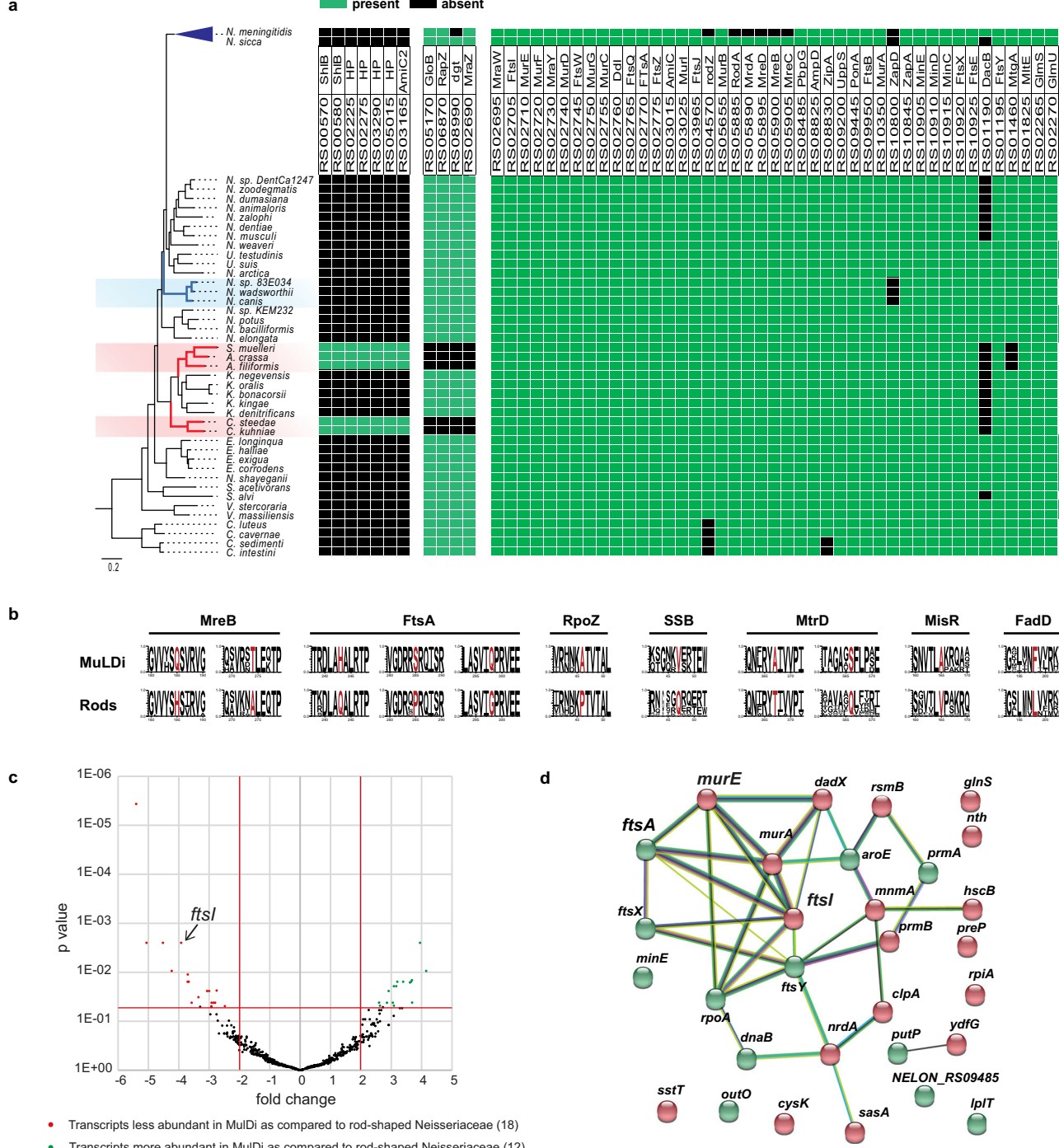

**Fig. 5 | Comparative genomics and transcriptomic of rod-shaped and MuLDi Neisseriaceae. a** Phylogenetic tree of *Neisseriaceae* species (left) and table displaying the distribution, within the family, of putatively inserted (left) or deleted (middle) genes. In addition, selected genes known to be involved in cell growth and/or division are shown (right). Individual genes were considered to be present when they had a sequence similarity ≥60% relative to *N. elongata* [an e-value cut-off of 1e⁻¹⁰ has also been applied in TBLASTN version 2.7.1 (Altschul et al.[102]). Present genes are indicated with *S. muelleri* locus_tag (such as RS00570 for BWP33_RS00570). All other genes are indicated with *N. elongata* locus_tag (such as RS02740 for NELON_RS02740). The putative encoded protein associated with each gene are also specified. The green and black squares indicate genes that are present

or absent, respectively. HP hypothetical protein. **b** Weblogo of the amino acid sequences, of the 7 proteins displaying amino acid permutations rod-shaped or for MuLDi, detected with amino acid permutations between rod-shaped and MuLDi *Neisseriaceae*. **c** Volcano-plot: *p* value is plotted against fold change calculated using DeSeq2. Red and green points correspond to transcripts that are less or more abundant in MuLDi as compared to rod-shaped *Neisseriaceae*, respectively. Source data and statistics are provided as a Source Data file. **d** STRING association analysis. *ftsA*, *ftsI* and *murE* from the *dcw* cluster are highlighted. In red are transcripts that are less abundant in MuLDi *Neisseriaceae* and in green are transcripts that are more abundant in MuLDi as compared to rod-shaped *Neisseriaceae*.

37 *Neisseriaceae* species (core proteome) (Fig. 5b). Strikingly, we detected amino acid permutations in only seven out of the 438 proteins (1.6%). Namely, three and two permutations were found in FtsA and MreB, respectively, two proteins which are both involved in bacterial morphogenesis[51]. Consistently, the phylogeny based on FtsA or MreB protein sequences (Supplementary Fig. 10) revealed that all the MuLDi sequences clustered together, suggesting convergent evolution of these proteins or horizontal gene transfer between MuLDi species. In addition to FtsA and MreB, we also found two permutations in the efflux pump membrane transporter MtrD, and one permutation in the DNA-directed RNA polymerase subunit RpoZ, the single stranded DNA-binding protein Ssb, the two-component regulator MisR and the long-chain-fatty-acid-CoA ligase FadD.

Altogether, comparative genomics of rod-shaped versus MuLDi *Neisseriaceae* identified 18 genetic loci whose presence, absence or mutation strictly correlate with the rod-to-MuLDi transition. Notably, these genetic loci include *amiC2*, encoding for a cell wall amidase, the *dcw* cluster regulator-encoding gene *mraZ* and the actin homolog-encoding gene *mreB*.

### Downregulation of *dcw* cluster genes in MuLDi *Neisseriaceae*

As several genes encoding for regulators were mutated in MuLDi *Neisseriaceae*, we employed RNAseq to determine differential gene expression patterns between MuLDi ($n = 5$) and rod-shaped ($n = 5$) *Neisseriaceae* cultured in the same conditions (GCB agar Media, 6 h, 37 °C 5% $CO_2$). To compare gene expression between species, we standardized the annotation of the five rod-shaped and the five MuLDi *Neisseriaceae* genomes by inferring gene orthology using BlastP (55% of similarity as a cut-off). Using the NetworkX python programming package[52], we reannotated clusters of homologous genes in each genome (for example, the *ftsZ* gene will be called NEISS_1241 in all genomes). By doing so, we could count the reads that mapped to each gene in each species and perform DESeq2 statistical analyses using the core transcriptome. Strikingly, our analysis (Fig. 5c, d) showed that the majority of the significantly differentially regulated genes are involved in cell envelope synthesis (as demonstrated by their clustering in the String analysis shown in Fig. 5d). Namely, 12 genes appeared upregulated in MuLDi species, including *minE, ftsA, ftsX* and *ftsY* involved in cell division. More importantly, the 19 downregulated genes in MuLDi species included *murE* and *ftsI*, which are part of the *dcw* cluster.

To conclude, based on comparative interspecies RNA-seq, the absence of *mraZ* correlates with a downregulation of the *dcw* cluster (including *ftsI*) in MuLDi *Neisseriaceae*.

### Downregulation of *dcw* cluster genes in *N. elongata mraZ* deletion mutants

To test whether deletion of *mraZ* in the rod-shaped *Neisseriaceae N. elongata* could cause downregulation of *dcw* cluster genes (consistent with the apparent downregulation of the *dcw* cluster in MuLDi *Neisseriaceae* that naturally lost *mraZ*, see previous section), we compared the transcriptomes of wild-type *N. elongata* and a *mraZ* deletion mutant thereof. This revealed that five genes located downstream of *mraZ* (*mraW, ftsL, ftsI, murE* and *murF*) were downregulated (Fig. 6a, b). These results were confirmed by quantitative real-time PCR (Fig. 6c). Moreover, overexpressing *mraZ* (by inserting it, ectopically, downstream of the *nrq* locus in the *N. elongata* Δ*mraZ* mutant) led to overexpression of the first seven genes of the *dcw* cluster (Fig. 6a–c). Although the *N. elongata* Δ*mraZ* mutant did not display significant morphological defects (Fig. 6d), *N. elongata* overexpressing *mraZ* under the *porB* promoter (Δ*mraZ porBp-mraZ*) were smaller (Fig. 6d, e).

Collectively, we showed that *mraZ* is regulating transcription of the first five genes of the *N. elongata dcw* cluster and that expression of these genes impacts *N. elongata* cell length.

### Recapitulation of MuLDi-specific genetic changes in the rod-shaped *Neisseriaceae*, *N. elongata*, resulted in longer septa

After deleting *mraZ*, we tested whether individual changes at other MuLDi-specific loci could turn the rod-shaped *Neisseriaceae N. elongata* in a MuLDi bacterium. Deletion of *dgt, gloB*, or *rapZ* did not change *N. elongata* morphology (Supplementary Fig. 11a). All the same, introduction of *amiC2* (along with its neighboring gene *cdsA*) in *N. elongata* did not result in significant shape or growth anomalies (Fig. 7a). However, the allelic exchange of *N. elongata mreB* with *S. muelleri mreB* resulted in significantly longer cells (Fig. 7a and Supplementary Fig. 11b, c).

In a final attempt to turn the rod-shaped *N. elongata* into a MuLDi *Neisseriaceae*, we used an unmarked deletion-based technique developed by us[15] to concomitantly delete *dgt, gloB, mraZ* and *rapZ*, substitute *N. elongata mreB* with *S. muelleri mreB* and introduce *amiC2/cdsA*. As shown in Fig. 7a and Supplementary Fig. 11b, c, N. *elongata* Δ*dgt*, Δ*gloB*, Δ*mraZ*, Δ*rapZ* with *mreB_sm* were longer and branched. More importantly, the substitution of *mreB_ne* with *mreB_sm* together with the introduction of *amiC2/cdsA* resulted in cells with a longer septum and shorter axis perpendicular to the septum (Fig. 7b–d). Namely, the ratio between the two cell axes changed from $0.61 \pm 0.25$ ($n = 186$), in the wild-type, to $0.95 \pm 0.29$ ($n = 174$) in the mutant *N. elongata*.

All in all, even if our attempt to genetically manipulate the rod-shaped *N. elongata* into a MuLDi bacterium did not result into a complete transverse-to-longitudinal division switch (ratio between the two cell axes >1), the observed increase in septum length suggests that the genetic events identified by comparative genomics have participated in the rod-to-MuLDi *Neisseriaceae* transition.

## Discussion

There is a huge discrepancy between the number of known prokaryotic species and the number that have been characterized morphologically. This makes it hard to predict how the shape and the growth mode of bacteria evolved. In an attempt to fill this knowledge gap, we focused on MuLDi *Neisseriaceae* occurring in the oral cavity of warm-blooded vertebrates, including humans. Whole genome-based phylogenetic analysis, coupled with ultrastructural analysis, indicated that MuLDi bacteria evolved from a rod-shaped ancestor. Although rod-shaped *Neisseriaceae* septate transversally, our incubations with a set of fluorescently labeled PG precursors showed that MuLDi *Neisseriaceae* septate longitudinally – in *A. filiformis* in a distal-to-proximal fashion, in *S. muelleri* and *C. steedae* synchronously, from both poles to midcell (notably, the other two known species of the *Alysiella* and *Conchiformibius* genera, *A. crassa* and *C. kuhniae*, also septate longitudinally, the former unidirectionally and the latter bidirectionally; Supplementary Figs. 6e and 8c, respectively). Furthermore, we observed that in these bacteria, new PG was not inserted concentrically, but as a medial sheet guillotining each cell. Finally, full-scale comparative genomics revealed MuLDi-specific differences that set them apart from rod-shaped members of the *Neisseriaceae* (e.g., *amiC2* acquisition, *mraZ* loss and amino acid changes in the cytoskeletal proteins MreB and FtsA). Supporting the role of specific genetic changes in the rod-to-MuLDi transition, introduction of *amiC2* and allelic substitution of *mreB* in the rod-shaped *Neisseriaceae N. elongata* resulted in cells with longer septa. Taken together, we presented two novel modes of septal growth and we identified genetic events that likely contributed to the evolution of bacterial multicellularity, longitudinal division and, possibly, polarization in a group of mammalian symbionts.

Multiple phylogenetic studies have suggested that the wide palette of bacterial morphotypes we observe today evolved from rod-shaped bacteria, which makes us consider their shape as the ancestral one[53,54]. Our genome-based phylogenetic reconstruction revealed that also MuLDi *Neisseriaceae* evolved from an ancestral rod-shaped

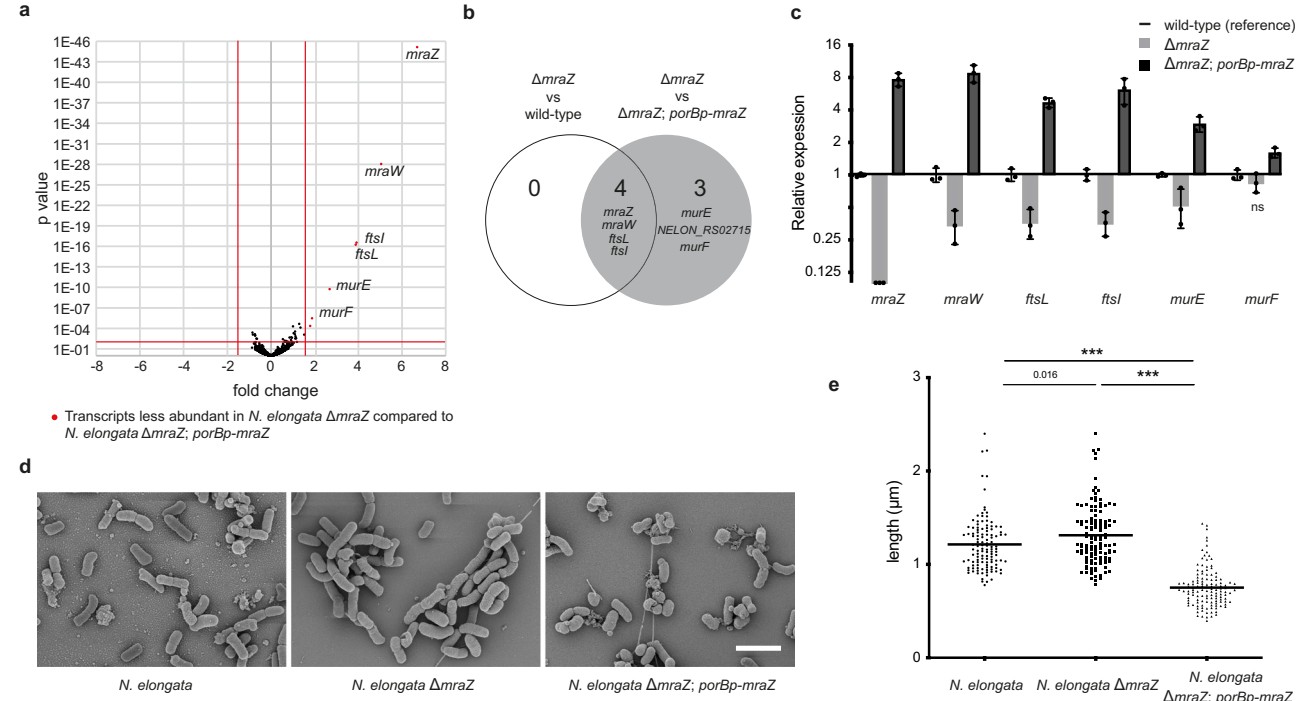

**Fig. 6 | Downregulation of the *dcw* cluster in *N. elongata* Δ*mraZ*. a** Volcano plot of RNAseq analysis of an *N. elongata* Δ*mraZ* and complemented. *p* value is plotted against fold change and were calculated using DeSeq2. Red points represent genes upregulated in MraZ-overexpressing *N. elongata* (Δ*mraZ; porBp-mraZ* – i.e., *mraZ* under the control of the strong and constitutive *porB* promoter), as compared to *N. elongata* Δ*mraZ*. **b** Venn diagram showing genes (*mraZ, mraW, ftsL and ftsI*) upregulated in *N. elongata* wild-type as compared to *N. elongata* Δ*mraZ*. **c** Transcript abundance of *dcw* cluster genes measured by qRT-PCR in *N. elongata* expressing or not expressing MraZ. Data represent mean (*n* = 3 biologically independent samples ± SD) and are representative of three independent experiments. Statistical test used was Unpaired *t* test with Welch's correction by comparing ΔmraZ to the parental wild-type and the Δ*mraZ; porBp-mraZ* to the parental Δ*mraZ* (ns not significant). **d** Scanning electron microscopy of *N. elongata* expressing or not expressing MraZ. Scale bar is 2 μm. **e** Median cell length measurements of *N. elongata* expressing or not expressing MraZ (*n* = 120 biologically independent cells). Data are presented with the median and are representative of at least two independent experiments. Statistical test used was One-way ANOVA, with Bonferroni's multiple comparisons test (***$p < 0.001$). Source data and statistics are provided as a Source Data file (for **a**, **c** and **e**).

bacterium. It remains uncertain whether these two MuLDi lineages evolved independently but convergently or whether species belonging to the genus *Kingella* also evolved the MuLDi phenotype, but subsequently reverted to transverse division. This could be resolved in the future by isolation of more species closely related to MuLDi or *Kingella* spp. As for what makes *Conchiformibius* and *Simonsiella* divide from both poles and *Alysiella* from one pole only, comparison of their genomes and transcriptomes is needed to decipher the underlying mechanisms. Irrespective of differences in the directionality of cell wall construction, we speculate that the MuLDi phenotype may have favored colonization of (or nutrient uptake from) the buccal cavity, which is characterized by rapidly epithelial cells shedding and salivary flow[55]. Indeed, multicellularity makes cooperation between cells possible, for example in the form of division of labor, and may therefore help bacteria to survive nutritional stress (see for example ref. 56). Although previous morphological studies suggested that the terminal cells of *S. muelleri*[10,17] and *C. steedae*[11] might phenotypically differ from the central ones and although we observed cells with thicker PG every 14 cells in *C. steedae* (Supplementary Figs. 2e, 8a and Supplementary Movie 7), future studies are needed to clarify whether different cell types exist within each filament.

Multicellularity may arise via three distinct processes: (1) aggregation of individual cells resembling the initial stages of biofilm formation[57]; (2) the formation of syncytial filaments via crosswalls segmenting the mother cell, but not separating it into daughter cells (streptomycetes[58]); and (3) incomplete cell fission after cell division to produce chains of cells (referred to as clustered growth, e.g., filamentous cyanobacteria[59]). TEM analysis of MuLDi symbionts revealed

that these *Neisseriaceae* share their lateral PG and are surrounded by a common outer membrane which makes them resemble to cyanobacteria. If MuLDi cells belonging to the same filament appear to be synchronized (Figs. 3 and 4; Supplementary Figs. 6 and 7), additional studies are needed to find out whether division of labor occurs among cells belonging to the same filament and whether their cytoplasms are connected by septal junctions and/or hemidesmosomes[59]. As for the mechanism underlying MuLDi cell septation, ultrastructural analysis suggests that, after a first round in which PG is synthesized and the CM invaginates, a second round occurs where the PG layer is split into two, concomitantly with the invagination of the outer membrane until midcell (Supplementary Fig. 3).

Although longitudinal septation is clearly not a prerequisite of bacterial multicellularity (here defined as clusters of at least three cells), these two phenotypic traits appeared to have evolved jointly in the *Neisseriaceae*. Longitudinal septation has also been shown in the nematode symbionts *Candidatus Thiosymbion oneisti* and *T. hypermnestrae*[20,21], as well as in the fruit fly endosymbiont *Spiroplasma poulsonii*[22]. In these three unicellular symbionts, the tubulin homolog FtsZ is localized at the septal plane and is therefore thought to mediate septal PG insertion. As for the actin homolog MreB, it was shown to form a medial ring-like structure in *Ca*. Thiosymbion throughout the cell cycle and to be required for septal FtsZ localization and PG insertion[38]. Indeed, its pharmacological inactivation impaired both *Ca*. Thiosymbion growth and division[38]. Although the localization pattern of MreB in MuLDi *Neisseriaceae* is currently unknown, (1) its presence in their genomes, (2) its transcriptional expression, (3) the identification of two MuLDi-specific amino acid permutations (H185Q and

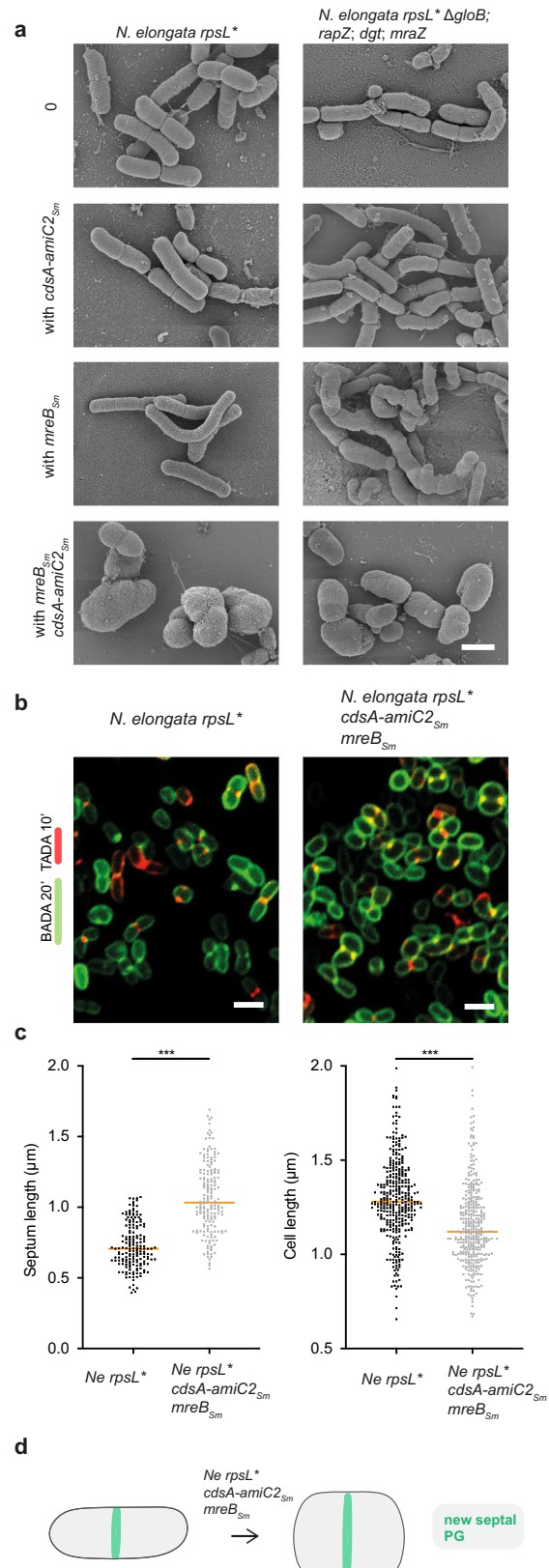

**Fig. 7 | Recapitulation of MuLDi-specific genetic changes in the rod-shaped *Neisseriaceae N. elongata*. a** Scanning Electron Microscopy of *N. elongata* (*rpsL\**) wild-type (left panels) or harboring multiple deletions (Δ*dgt*, Δ*gloB*, Δ*mraZ*, Δ*rapZ*, right panels), with or without the *mreB_Ne*/*merB_Sm* allelic exchange, with or without the addition of *cdsA-amiC2*. The results are representative of at least three independent analyses. **b** *N. elongata* (*rpsL\**) wild-type (left) or harboring the *mreB_Ne*/*merB_Sm* allelic exchange and *cdsA-amiC2* (right) and **c** median length of the septum (*n* = 170 biologically independent cells) and of the cell axis perpendicular to the septum (*n* = 340 biologically independent cells) in *N. elongata* (*rpsL\**), wild-type (left) or harboring the *mreB_Ne*/*merB_Sm* allelic exchange and *cdsA-amiC2* (right). Data are presented with the median and are representative of at least two independent experiments. Statistical test used was Unpaired Two-Tailed *T* test (\*\*\**p* < 0.001). Source data and statistics are provided as a Source Data file. **d** Schematic representation of a septating wild-type (*rpsL\**) *N. elongata* (left) and of *N. elongata* harboring the *mreB_Ne*/*merB_Sm* allelic exchange and *cdsA-amiC2* (right). Scale bar is 1 μm.

in longitudinally dividing *Ca.* Thiosymbion when compared to *E. coli* (S185N)[33]. It should also be noted that the effect of the allelic substitution of MreB on the morphology of *N. elongata* depended on the genetic background (i.e., presence of MuLDi-specific genes and/or absence of rod shape-specific genes). This, in addition to the pleiotropic effect of MreB reported in other studies[28], can make this protein accountable for accommodating multiple cell shape adaptations (e.g., rod-to-coccus, rod-to-MuLDi). If we still do not know whether MreB and/or FtsZ place the insertion of the PG synthesis machinery at the septum, based on our confocal-based 3D reconstructions, new septal PG is not inserted in successive, concentric rings or ellipses, as shown for model rods[39,40,60] and nematode symbionts[38], respectively.

In addition to MreB amino acid changes, MuLDi-specific loss of *mraZ* led to the misregulation of the *dcw* cluster. MraZ has been described as a highly conserved transcriptional regulator of the *dcw* cluster, of which *mraZ* is the first gene[45,46]. The *dcw* cluster is a group of genes involved in the synthesis of PG precursors and cell division[61] that is conserved in most bacterial genomes[62–64]. Throughout the *Neisseriaceae*, the *dcw* cluster consists of 14–16 tightly packed genes in the same orientation and mostly in the same order, with *midA* located before the cluster in reverse orientation (Supplementary Fig. 9). The fact that, in the *Neisseriaceae*, the gene content and orientation of the *dcw* cluster mostly mirrored the phylogenetic placement of each species, suggests that the *dcw* cluster evolved vertically. Moreover, having a fragmented *dcw* cluster (as in the case of MuLDi *Neisseriaceae* and some *Kingella* species) does not seem to impact cell morphology, given that both rod-shaped and MuLDi species may bear or not bear split *dcw* clusters. Of note, in spite of fragmentation, bacteria can retain some gene sub-clusters (e.g., "*mraW-ftsL, ftsI, murE* and *murF*", "*ftsW, murG*" and "*murC, ddl, ftsQ, ftsA, ftsZ*"), probably because the genes grouped in a given sub-cluster need to be co-transcribed (Mingorance et al.[65]). If several studies agree on the regulatory role of MraZ on the *dcw* cluster expression[45,46,66,67], a lot remains to be done to understand the details of this regulation (e.g., what triggers MraZ activity). In this study, *dcw* cluster upregulation in MraZ-overexpressing *N. elongata* led to shorter cells. This agrees with observing the opposite phenotype (filamentation) in *E. coli* when the *dcw* cluster was downregulated[45]. Altogether, these studies suggest that MraZ controls cell division rate by regulating the *dcw* cluster and we speculate that, in *mraZ*-less MuLDi *Neisseriaceae*, its absence may have altered the balance between the divisome and the elongasome machineries (i.e, the elongasome might contribute to PG synthesis at the septum).

Finally, comparative genomics highlighted the importance of the acquisition of the *cdsA/amiC2* locus in MuLDi *Neisseriaceae*. Although its sole addition in *N. elongata* does not result in morphological changes, when we combined it with the allelic substitution of *mreB_sm*, we observed cells with longer septa. This suggests that the AmiC2 amidase may regulate MuLDi septation. Intriguingly, HPLC analyses of PG extracted from 17 rod-shaped bacteria and from three MuLDi

T247A), and (4) the fact that introducing a MuLDi MreB in the rod-shaped *N. elongata* (with the concomitant insertion of *amiC2* or deletion of *dgt, gloB, mraZ, rapZ*) led to shape aberrations suggest that MreB is involved in PG insertion in MuLDi *Neisseriaceae*. Intriguingly, amino acid 185, located after the GVVYS motif, is substituted in MuLDi MreBs when compared to rod-shaped *Neisseriaceae* (H185Q), but also

*Neisseriaceae* (*A. filiformis*, *S. muelleri* and *C. steedae*) showed that MuLDi PG is richer in M44 (Supplementary Fig. 5e), suggesting higher amidase activity in these *Neisseriaceae*. Concerning the *amiC2*-associated genetic locus *cdsA*, it encodes for a phosphatidate cytidylyltransferase putatively implicated in phospholipid biosynthesis. Given that the presence of anionic phospholipids (cardiolipin and phosphatidylglycerol) has been shown to repel MreB[68], we can hypothesize that CdsA affects the composition of the membrane and, therefore, the localization of MreB.

Despite all our efforts, we could not turn the rod-shaped *N. elongata* into a complete MuLDi *Neisseriaceae* even upon, concomitantly, replacing MreB, inserting *amiC2/cdsA* and deleting *dgt*, *gloB*, *mraZ* and *rapZ*. This could be because we could not recreate all genetic events (such as replacing $ftsA_{ne}$ with $ftsA_{sm}$ due to its proximity to *ftsZ*), or it could be due to the existence of other undetected events (such as species-specific events that resulted in a convergent phenotype), or finally, to prior events such as those that are also present in rod-shaped *Kingella* spp. (see Supplementary Data 5) but that do not cause the MuLDi phenotype.

How could rod-shaped, transversally dividing bacteria evolve into longitudinally dividing ones? Permanent cell shape transitions may have resulted from modifications (e.g., gene deletions, insertions and nucleotide polymorphisms) of genetic loci involved in morphogenesis (e.g., *mreB*, *amiC2*) and, additionally, in those involved in their transcriptional regulation (e.g., *mraZ*). Two evolutionary scenarios were proposed[33,34,38]: (1) an ancestral rod was compressed by its poles so that it got shorter and fatter, or (2) an ancestral rod rotated its septation axis by 90 degrees. Our results suggest that, in the course of evolution, the cell width of an ancestral rod increased (and its length decreased), perhaps following a misbalance between elongation and division. However, genetic tools are needed to gain insights on MuLDi *Neisseriaceae* evolution by, for example, visualizing the localization pattern of FtsZ and MreB or by attempting reversion into unicellular, and possibly, into transversally dividing bacteria such as *N. elongata*.

To date, most protein function studies have been conducted in either pathogenic or bacterial species that are easy to culture and manipulate in the laboratory such as *E. coli* and *B. subtilis*. In addition to these models, efforts to study other morphologies including commensal species are necessary to understand bacterial cell evolution, but also to increase the pool of protein targets (e.g., antibiotic targets) for industrial and biopharmaceutical applications. Throughout their evolution, *Neisseriaceae* succeeded in repeatedly, and seemingly effortlessly, evolve different cell shapes (e.g., coccoid, MuLDi). Moreover, they are the only known multicellular longitudinally dividing bacteria that may thrive in humans, but which are also cultivable and, likely, genetically tractable. We hence propose the use of *Neisseriaceae* as models to study how longitudinal division and multicellularity evolved, as well as the molecular and cell biological mechanisms underlying the establishment of bacterium-animal symbioses.

## Methods

### Bacterial strains and culture conditions
The bacterial strains *Neisseria elongata* subsp. *elongata* (DSM 17712), *Alysiella filiformis* (DSM 16848), *Simonsiella muelleri* (DSM 2579), and *Conchiformibius steedae* (DSM 2580) were obtained from the German Collection of Microorganisms and Cell Cultures GmbH (DSMZ). *Neisseria elongata* subsp. *glycolytica* (ATCC 29315) and *Simonsiella muelleri* (ATCC 29453) were obtained from the American Type Culture Collection (ATCC). *N.* sp. DentCa1/247 was a gift from Dr. Nathan Weyand (U. of Ohio). For FDAA incubations, western blots, immunostaining and membrane staining, we used BSTSY (*N. elongata*, *C. steedae*), PY (*A. filiformis*), or meat extract (*S. muelleri*) agar plates that were incubated overnight at 37 °C. For BSTSY, PY and meat extract media composition please refer to Supplementary Table 1. For all other experiments, bacteria were streaked from −70 °C freezer stocks onto Gonococcal

culture media supplemented with Kelloggs supplement (GCB) and grown overnight at 37 °C in 5% $CO_2$ incubator. Single colonies were subcultured into the respective liquid media with agitation at 120 rpm and grown to exponential phase ($OD_{600}$ 0.1–0.6). For cloning experiments *E. coli* DH5α cells were cultured onto Lysogeny broth (LB) media at 37 °C. When required, antibiotics were used as follows: kanamycin (50 μg/ml for *E. coli*; 100 μg/ml for *N. elongata*), erythromycin (300 μg/ml for *E. coli*; 3 μg/ml for *N. elongata*), chloramphenicol (25 μg/ml for *E. coli*; 5 μg/ml for *N. elongata*), and streptomycin (100 μg/ml for *N. elongata*). Transformation of *N. elongata* was done using linearized plasmid or PCR product by dropping ~500 ng of DNA on fresh cultures on GCB media supplemented with 10 mM $MgCl_2$ and incubated for 6–12 h before subculturing on GCB media containing the appropriate antibiotics and Xgal if needed as described previously[16].

### Time-lapse imaging of *N. elongata*
Strains were streaked from −70 °C freezer stocks onto BSTSY agar plates and grown overnight at 37 °C with 5% $CO_2$. Single colonies were transferred to liquid culture and grown to exponential phase ($OD_{600}$ 0.2). Cells were spotted onto pads made of 0.8% SeaKem LE Agarose (Lonza, Cat. No. 50000) in BSTSY and topped with a glass coverslip. Cells were transferred to an Okolab stage top chamber to control temperature (37 °C) and gas ($CO_2$ 5% and $O_2$ 18%). Images were recorded with inverted Nikon Ti-2 microscopes using a Plan Apo 100 × 1.40 NA oil Ph3 DM objective using Hamamatsu Orca FLASH 4 camera. Images were processed with NIS Elements 5.02.01 software (Nikon). In all experiments, multiple x/y positions were imaged. Representative images were processed using the Fiji 2.1.0/1.53c software package.

### Time-lapse imaging of *A. filiformis*, *S. muelleri* and *C. steedae*
Strains were streaked from −70 °C freezer stocks onto PY (*A. filiformis*), meat extract (*S. muelleri*) or BSTSY (*C. steedae*) agar plates grown overnight at 37 °C with 5% $CO_2$. Single colonies were transferred to liquid culture and grown to exponential phase ($OD_{600}$ 0.2–0.5) at 37 °C shaking at 180 rpm agitation. For all strains, 250 μL of diluted exponential phase cultures (OD 0.025) were loaded into the cell loading well of a prepared (shipping solution removed and washed three times with sterile appropriate media) B04A-03 microfluidic plate (Merck-Millipore). Time-lapse imaging was performed using CellASIC® ONIX Microfuidic System. The ONIX manifold was sealed to the B04A-03 plate. CellASIC® ONIX2 System was used as the microfluidics control software. First, a flow program was set up to prime flow channel and culture chamber by flowing medium from inlet wells 1–5 at 34.5 kPa for 2 min. Second, cells were loaded onto the plate at 13.8 kPa for 15 s. Priming run was performed for 5 min with pressure set to 34.5 kPa. The medium flow was set at 12 kPa throughout the experiment for 12 h with sterile appropriate media. Images were recorded with an inverted Nikon Ti-E microscope using a Plan Apo 60XA oil Ph3 DM objective using Hamamatsu Orca FLASH 4 camera. Images were processed with NIS Elements 5.02.01 software (Nikon). In all experiments, multiple x/y positions were imaged. Representative images were processed using the Fiji 2.1.0/1.53c software package.

### Electron microscopy
For transmission electron microscopy, half a loopful of 6–8 h old bacterial cultures were fixed by direct resuspension in 500 μl of 2.5% glutaraldehyde in 0.1 M cacodylate buffer and incubated for at least 1 h at 4 °C. Cells were then pelleted through centrifugation at 3075 × *g* for 3 min and washed 3 times in 500 μl 0.2 M cacodylate wash buffer solution (pH 7.2). 30–50 μl of wash solution containing bacterial cells was pipetted onto Formvar Carbon 200 mesh copper grids (Sigma-Aldrich) and negative staining done using 1% phosphotungstic acid (PTA) for 2 s before imaging at the INRS-CAFSB platform using a Hitachi H-7100 electron microscope with AMT Image Capture Engine (version 600.147).

For scanning electron microscopy, fresh bacterial cells were cultured for 6 h in liquid media containing poly-L-Lysine (Sigma) coated glass slides. Cells were fixed using 2.5% glutaraldehyde in 0.1 M cacodylate buffer for 1 h at 4 °C then rinsed 3 times in 0.2 M cacodylate wash buffer solution (pH 7.2). Post fixation was subsequently done using 1% osmium tetroxide (in 0.2 M cacodylate) before gradual dehydration through increasing ethanol concentrations (25%, 50%,75%, 95% and 100%). Carbon dioxide critical point drying (CPD) and gold sputtering were done on Leica EM CPD300 and Leica EM ACE600 instruments respectively. The imaging was done at the electron Imaging Facility (Faculty of dental medicine, Université de Montréal, Québec, Canada) using a Hitachi Regulas 8220 electron microscope with the SEM operation software Regulus 8200 series.

## Peptidoglycan extraction and analysis

Peptidoglycan (PG) extraction was performed as previously described[16]. Bacterial cultures were harvested from solid agar plates using inoculation loops and emulsified in 10 ml of distilled water, the suspension mix was added drop by drop into 10 ml of 8% boiling sodium dodecyl sulfate (SDS) and boiled for and extra hour. After overnight storage at room temperature, the cells were washed six times using distilled water (pH 6.0) through ultracentrifugation at $39,000 \times g$ for 30 min. The final pellet was lyophilized and resuspended in distilled water (concentration 6 mg/ml or more) and stored at −20 °C until further use. Analysis of the muropeptide composition was performed essentially as described previously[69]. Samples were treated with Proteinase K (20 µg/mL, 1 h, 37 °C). The reaction was heat-inactivated and sacculi were further washed by ultracentrifugation. Finally, samples were digested overnight with muramidase (100 µg/mL) at 37 °C. Muramidase digestion was stopped by boiling and coagulated proteins were removed by centrifugation ($3075 \times g$, 15 min). For sample reduction, the pH of the supernatants was adjusted to pH 8.5–9.0 with sodium borate buffer and sodium borohydride was added to a final concentration of 10 mg/mL. After incubating for 30 min at room temperature, pH was adjusted to 3.5 with orthophosphoric acid. The soluble muropeptides were analyzed by high-performance liquid chromatography (HPLC; Waters Corporation, USA) on a Kinetex C18 column (150 × 4.6 mm; 2.6 µm particle size, 100 Å) (Phenomenex, USA) and detected at 204 nm with UV detector (2489 UV/Visible, Waters Corporation, USA). Muropeptides were separated with organic buffers at 45 °C using a linear gradient from buffer A (formic acid 0.1% (v/v) in water) to buffer B (formic acid 0.1% (v/v) in 40% acetonitrile) in an 18-min-long run with a 1 ml/min flow. Quantification of muropeptides was based on their relative abundances (relative area of the corresponding peak) normalized to their molar ratio. The molar percentage was calculated for each muropeptide. This relative molarity was also used to calculate the molar percentage of crosslinked muropeptides. The Waters Empower 3, build 3471 software (Waters Corporation, USA) was used for acquiring and managing the chromatographic information. Muropeptide identity was confirmed by MS analysis, using a UPLC–MS (UPLC system interfaced with a Xevo G2/XS Q-TOF mass spectrometer from Waters Corporation, USA). Data acquisition and processing was performed using UNIFI software platform (Waters Corporation, USA).

## FDAA incubations

To sequentially label cells with HADA (7-hydroxycoumarin-3-carboxylic acid-D-alanine, blue), BADA (BODIPY FL-D-alanine; green) and TADA (TAMRA-D-alanine; red), all three provided by Michael van-Nieuwenhze, exponential phase cells were pelleted, resuspended in medium containing the first label and then grown at 37 °C. Media composition is described in Supplementary Table 1, and incubation times and order for each *Neisseriaceae* species are listed in

Supplementary Table 2, respectively. After the first interval cells were washed twice with fresh medium (37 °C) and centrifuged between washes ($7000 \times g$ for 2 min at RT). After this, the cell pellets were resuspended in pre-warmed medium containing label two. For triple labeling, cells were washed twice and resuspended in medium containing the third label. Cells were then immediately treated with 70% ice-cold ethanol and incubated on ice for 1 h. Ethanol-fixed cells were collected via centrifugation ($7000 \times g$ for 2 min at RT), washed twice with 4 °C 1 × Phosphate Buffered Saline (PBS, pH 7.4), resuspended in PBS, and stored on ice before imaging.

## EDA-DA incubation and click-chemistry

To track symbiont cell wall growth followed by immunolabeling *A. filiformis* cells were grown over night on PY plates. Single colonies were incubated in 10 mM ethynyl-D-alanyl-D-alanine (EDA-DA, a D-amino acid carrying a clickable ethynyl group) for 30 min, resuspended in pre-warmed PY medium, washed twice ($7000 \times g$ for 2 min at RT) and treated with 70% ethanol like described before. After that, cells were rehydrated and washed in PBS containing 0.1% Tween 20 (PBT). Blocking was carried out for 30 min in PBS containing 0.1% Tween 20 (PBT) and 2% (wt/vol) bovine serum albumin (blocking solution) at room temperature. An Alexa488 fluorophore was covalently bound to EDA-DA via copper catalyzed click-chemistry by following the user manual protocol for the Click-iT reaction cocktail (Click-iT EdU Imaging Kit, Invitrogen). The cells were incubated with the Click-iT reaction cocktail for 30 min at RT in the dark. Unbound dye was removed by a 10-min wash in PBT and one wash in PBS. For immunostaining of clicked bacterial cells, cells were washed for 10 min in PBT and subsequently incubated with blocking solution for 30 min at room temperature in the dark. From here on, immunostaining was performed as described below.

## Western blots

Proteins from bacteria cells were separated by reduced sodium dodecyl sulfate (SDS)-polyacrylamide gel electrophoresis (PAGE) on NuPAGE 4%–12% Bis-Tris pre-cast MOPS gel (Invitrogen), respectively, and each blotted onto Hybond ECL nitrocellulose membranes (Amersham Biosciences). Membranes were blocked for 45 min in PBS containing 5% (wt/vol) nonfat milk (PBSM) at room temperature and incubated overnight at 4 °C with a 1:1,000 dilution of sheep polyclonal anti-*E. coli* K88 fimbrial protein AB/FaeG antibody (ab35292, Abcam) in PBSM. For the negative control, the primary antibody was omitted. After five 6 min-long washes in PBSM and one final wash in PBS containing 0.1% Tween20, the blot was incubated for 1 h at room temperature with a horseradish peroxidase-conjugated anti-sheep secondary antibody (1:10,000; Amersham Biosciences) in PBSM. Protein-antibody complexes were visualized using ECL Plus detection reagents (Amersham Biosciences).

## Immunostaining

Exponential phase cells were fixed overnight in 3% formaldehyde at 4 °C. Cells were collected via centrifugation ($7000 \times g$ for 2 min at RT), washed twice with PBS and resuspended in PBS containing 0.1% Tween 20 (PBT). Blocking was carried out for 1 h in PBT containing 2% (wt/vol) bovine serum albumin (blocking solution) at room temperature. After that, cells were incubated with a 1:500 dilution of sheep polyclonal anti-*E. coli* K88 fimbrial protein AB/FaeG antibody (ab35292, Abcam) overnight at 4 °C in blocking solution. Upon incubation with primary antibody (or without in the case of the negative control) samples were washed three times in PBT and incubated with an Alexa555 conjugated anti-sheep antibody (Thermo Fisher Scientific) at 1:500 dilution in blocking solution for 1 h at room temperature. Unbound secondary antibody was removed by two washing steps one in PBT and one in PBS. Cell pellets were resuspended in PBS containing 5 µg/mL Hoechst for 20 min and subsequently washed and resuspended with PBS. 1 µL of

the bacterial solution was mixed with 0.5 μL of Vectashield mounting medium (Vector Labs) and mounted on an agarose slide.

## Nile red membrane staining

Exponential phase cells were fixed overnight in 2% formaldehyde at 4 °C. Cells were collected via centrifugation (7000 × g for 2 min at RT), washed twice with PBS and resuspended in PBS containing 10 μg/mL Nile Red (Stock is prepared with DMSO; ThermoFisher N1142) and 5 μg/mL Hoechst for 15 min in the dark at room temperature. Cells were washed and resuspended in PBS and subsequently 1 μL of the bacterial solution was mixed with 0.5 μL of Vectashield mounting medium (Vector Labs) and mounted on an agarose slide.

## Fluorescence microscopy

For Figs. 3a, b, 4a−c, Supplementary Figs. 4, 6a−c, and 7a−c immunostained or FDAA-labeled bacteria were imaged using a Nikon Eclipse NI-U microscope equipped with a MFCool camera (Jenoptik) and images were acquired using the ProgRes Capture Pro 2.8.8 software (Jenoptik). For Figs. 3d, 4e, Supplementary Figs. 6d, e, and 8a−c, FDAA-labeled bacteria were visualized with a Leica TCS SP8 X confocal laser scanning microscope. Images were taken with a 63X Plan-Apochromat glycerin objective with a NA of 1.30 and a refraction index of 1.46 (glass slide, glycerin and antifade mounting medium). The Leica software LASX (3.7.2.22383) including the Lightning deconvolution software package (Leica) was used for image acquisition and post-processing if necessary.

For Fig. 7b and Supplementary Fig. 11b, FDAA-labeled *N. elongata* wild type (*rpsL*\*) and mutant (*rpsL*\*; *cdsA-amiC*$_{SM}$; *mreB*$_{SM}$) were imaged at the INRS-CAFSB platform with a Zeiss LSM 780 AxioObserver confocal microscope equipped with a Zeiss Plan-Apochromat 100x/1.4 Oil M27. The Zeiss software Zen 2011 was used for image acquisition.

## FDAA fluorescence quantification and statistical analysis

Microscopic images were processed using the public domain software ImageJ 1.53k[70] in combination with plugin Fil-Tracer (this study). Cell outlines were traced and morphometric measurements recorded. Fluorescent intensities were measured along the septal plane and plotted as fraction of the normalized cell length. Automatic cell recognition was double-checked manually. For representative images, the background subtraction function of ImageJ 1.53k was used and brightness and contrast were adjusted for better visibility. Data analysis was performed using Excel 2021 (Microsoft Corporation, USA), plots were created with ggplot2 in R (http://www.R-project.org/). Septa length (Fig. 7 and Supplementary Fig. 11) of BADA and TADA labeled cells were analyzed using the public domain software Fiji[71]. Cell and septa lengths were measured manually. Notably, only cells that showed a BADA and TADA signal were considered for the septa length measurements. Two-tailed unpaired T tests were performed using GraphPad Prism version 9.3.0 for Mac (La Jolla California USA, www.graphpad.com). Figures were compiled using Adobe Photoshop 2021 and Adobe Illustrator 2021 (Adobe Systems, USA).

## Genome sequencing and assembly

Genomic DNA for WGS of *Neisseriaceae* species and PCR amplification of DNA used for cloning purposes or sequence verifications were extracted using Genomic Tip 20/G or 100/G kits (Qiagen) according to the manufacturer's instructions. The genome sequencing results are presented in Supplementary Data 1. Genomes were sequenced either using a Pacific Biosciences RS II system at the Génome Québec Innovation Centre (McGill University, Montréal, Canada) or using Oxford Nanopore technologies at the Bacterial Symbiont Evolution Lab (INRS, Laval, Canada). For PacBio, the reads were assembled de novo using HGAP v.4[72] available on SMRT Link v.7 (default parameters, except, min. subread length: 500; estimated genome size: 2.7 Mb). For

nanopore sequencing, DNA libraries were prepared following the Native barcoding genomic DNA procedure (with EXP-NBD104, EXP-NBD114, and SQK-LSK109) with MinIon MK1C device using the Min-KNOW 21.05.10 software. The base call was carried out using guppy_basecaller (version 5.0.11 + 2b6dbff) in sup mode. Reads were filtered by quality Q > 8 and separated by barcodes using guppy_barcoder (version 5.0.11 + 2b6dbff). The genome assembly was made by 3 programs: Canu (https://github.com/marbl/canu), Flye (https://github.com/fenderglass/Flye)[73] and Miniasm (https://github.com/lh3/miniasm)[74]. Then each ensemble was corrected in bases using Pilon (https://github.com/broadinstitute/pilon)[75]. Racon (https://github.com/isovic/racon)[76] Medaka (https://github.com/nanoporetech/medaka). All assemblies and assembly corrections were analyzed with Quast (https://github.com/ablab/quast)[77] and BUSCO (https://gitlab.com/ezlab/busco)[78]. The assembly with the least number of contigs and the greatest completeness was chosen.

## Core genome-based phylogeny of *Neisseriaceae*

All the *Neisseriaceae* genomes present on NCBI database at the time of the analyses were downloaded and the Average Nucleotide Identity (ANI) values were determined using GET_HOMOLOGUES version 20092018. Genome were grouped by their ANI > 96%. To simplify the analyses and to assure their quality (such as avoiding multiple contigs) a reference genome was selected in each group. A complete circular genome from a reference strain was preferred when possible (see Supplementary Data 1). All genomes were annotated with Prokka v1.14.5[79] providing the annotation files for further analysis. A nucleotide-level multifasta alignment of those genes included in the core-genome of *Neisseriaceae* family was performed with MAFFT by using the *-e -mafft* options within Roary v3.11.2[80]. A minimum percentage of 50% identity, and occurrence in 80% of the isolates was also considered by entering the *-i 50 -cd 80* options, respectively, to the Roary command line. Under these parameters, a total of 401 genes were finally included in the core-genome (see Supplementary Data 3). The resulting alignment was used for the subsequent phylogenetic analysis. Best evolutionary model was determined by ModelFinder within IQ-TREE version 1.6.3[81]. The best-fit model according to the Bayesian Information Criterion (BIC) was GTR + F + R10. Maximum-likelihood phylogenetic analysis was also performed with IQ-TREE[82] using 10,000 ultrafast bootstrap replicates[83]. The final phylogenetic tree was drawn with FigTree v1.4.4 (http://tree.bio.ed.ac.uk/software/figtree/) and rooted in *Crenobacter*. The results of this analyses are provided in Supplementary Data 6. The phylogenetic tree displayed in Fig. 1, and csv file associating genome name with morphology were used in PastML with default parameters to assess ancestral morphology. The prediction method was maximum-likelihood-MPPA (marginal posterior probabilities approximation), F81 model.

## Phylogenies of individual proteins

Individual phylogenies were performed for six proteins: N-acetylmuramoyl-alanine AmiC1 from *S. muelleri* ATCC 29453 (accession number AUX62143.1), and AmiC2 from *C. kuhniae* (accession number WP_027009548.1); division/cell wall cluster transcriptional repressor MraZ from *N. elongata* (accession number WP_204812527.1); RNase adapter RapZ from *N. elongata* (accession number WP_074896150.1); cell division protein FtsA from *Neisseria spp.* (accession number WP_003779891.1); and rod shape-determining protein MreB from *Neisseria spp.* (accession number WP_003747269.1). Protein sequences were searched against all the *Neisseriaceae* genomes included in the core-genome-based phylogeny, as well as in the complete bacterial repertoire found at NCBI. For this purpose, two separate databases were created: *Neisseriaceae*, including the genomes above mentioned; and *Bacterial*, which includes all the representative genomic sequences from the RefSeq database. The protein sequences of all the genomes in each of the two datasets were concatenated and the

protein databases created with the *makeblastdb* tool from BLAST version 2.6.0+. The resulting databases were used to investigate the presence of aforementioned proteins by BLASTP. Sequences with an e-value and similarity percentage greater than or equal to 1e$^{-10}$ and 50%, respectively, were retained for downstream analysis. Truncated proteins (i.e., split into contiguous coding sequences) were not considered to avoid artefacts in the clustering. Amino acid sequences of the hits obtained by BLASTP were retrieved from the entire set of genomes using faSomeRecords (https://github.com/santiagosnchez/faSomeRecords/blob/master/faSomeRecords.pl). The resulting multifasta were aligned with MAFFT v7[84], and maximum-likelihood phylogenetic analysis was performed using IQ-TREE using 1000 ultrafast bootstrap replicates. Evolutionary models were estimated with ModelFinder in IQ-TREE, and best-fits according to BIC were as follows: AmiC1 LG + F + I + G4; AmiC2 LG + R10; FtsA JTT + R4; MraZ LG + I + G4; MreB JTTDCMut+R3; RapZ LG + R4. Final trees were drawn with FigTree v1.4.4 and rooted in *Crenobacter*. The results of this analyses are provided in Supplementary Data 6.

## Genomic comparisons

For gene insertion and deletion, we have used the previously described MycoHIT pipeline[42,85]. We used complete genomes of all the rod-shaped and MuLDi *Neisseriaceae* species presented in Fig. 1. We excluded the second coccus lineage (*Neisseria wadsworthii*, *Neisseria canis* and *N.* sp. 83E034). We performed an alignment search with the standalone TBLASTN program[86] using the 2105 predicted proteins from *N. elongata* ATCC29315 or the 2349 predicted proteins from *S. muelleri* as the query sequences to search for matches in the genomic DNA of other organisms. We obtained two matrices of around 80,000 scores (2063 or 2105 protein sequences blasted against 37 genomes) providing two types of output: categorical (hit versus no hit) and quantitative (degree of similarity). To categorically assign that there was no hit, we employed the default E-value of e$^{-10}$. Thus, if the statistical significance ascribed to a comparison was greater than this E-value, we assigned a percentage of similarity of 0% to that comparison. To analyze quantitative results, we used MycoHIT[42] to assign absence of gene in all MuLDi *Neisseriaceae* and presence of the gene in all rod-shaped *Neisseriaceae* or vice versa. "Absence" was defined as lower values than 50% and "presence" as higher values than 55%.

Possible correlation between amino acid changes and cell shape was sought using CapriB[41]. Briefly, two databases were generated taking MuLDi *Simonsiella muelleri* ATCC 29453 (accession number GCA_002951835.1) and rod-shaped *Neisseria elongata* subsp. *glycolytica* ATCC 29315 (accession number GCA_000818035.1]) as references. The proteins encoded by each genome under study here were further compared against these two references by TBLASTN. Once the blast results were obtained, and the groups to be compared were defined, i.e., MuLDi versus rod-shaped, amino acid changes in proteins shared by both groups (identity threshold 60%) were investigated focusing on those amino acids conserved in the members of both groups but different between the two groups (I vs D option).

## RNA sequencing and analysis

Total RNA was extracted from 6 h cultures grown on GCB agar plates. The cells were harvested in RNA protect reagent (Qiagen). RNeasy Mini Kit (Qiagen) with RNase Free DNAse set (Qiagen) was used for RNA extraction according to the manufacturer's instructions.

The removal of ribosomal RNA for cDNA synthesis was done with NEBNext rRNA Depletion kit with 1 μg of total RNA in the purification using 1.8X Cytiva Sera-mag. For results presented in Fig. 6, the rRNA depleted mRNA were processed using the Illumina® Stranded mRNA Prep protocol without modification by Génome Québec Innovation Centre (McGill University, Montréal, Canada). 100 bp Pair-End Sequencing was performed with the NovaSeq 6000 system. FastQ

Reads have deposited on SRA database (PRJNA859935). Sequence reads were processed with FastQC (Version 0.73) to determine the quality before grooming by FastQ Groomer (Version 1.1.5). Paired FastQ reads were then aligned against *Neisseria elongata* subsp. glycolytica ATCC 29315 (accession number NZ_CP007726.1) genome using Bowtie2 (Version 2.4.2) and read counts were determined using htseq_count (Version 0.9.1) tool. Subsequently the gene expression of the transcripts was determined using DESeq2 (Version 2.22.40.6). Visualization of differentially expressed genes was done with Venn diagrams, drawn by a Web-based platform Venny 2.1 (https://bioinfogp.cnb.csic.es/tools/venny/).

For intra-genus transcriptomic comparison presented in Fig. 5, rRNA depleted mRNA were treated using the RevertAid RT Reverse Transcription Kit (K1691; Thermo Scientific™) with some adjustments. For first strand cDNA synthesis, 1 μl of random primer (3 μg/μL; 48190011; Invitrogen™) was added and the solution was incubated at 65 °C. For the second strand cDNA synthesis, procedure was followed without RNA removal step and by purifying the double-stranded cDNA with 1.8X Cytiva Sera-mag. The cDNA was eluted in 24 μL of nuclease-free water. Libraries were prepared by PCR BARCODING (96) AMPLICONS (SQK-LSK109) and PCR BARCODING (SQK-PBK004) (Oxford Nanopore technologies), as described by the manufacturer. FastQ Reads have deposited on SRA database (PRJNA859916).The base call was carried out using guppy_basecaller (version 5.0.11 + 2b6dbff) in sup mode, adapters were removed and filtered by quality Q > 8, they were separated by barcodes using guppy_barcoder (version 5.0.11 + 2b6dbff). In parallel, the ten indicated genomes were annotated with Prokka v1.14.5[79]. Using each of the protein sequences (.faa) files, a standalone BLASTP[87] was performed for each dyad possibility. Network connection was thereafter established with the python programming package NetworkX version 2.6.2[52] with a cut-off of 60% of similarity. Basically, all proteins showing more than 60% similarities with one of the members (putative homologs) were clustered together. Each cluster of proteins was named (example NEISS_1) and this name was used to replace the original locus-tags in the .GFF file (previously generated by Prokka). This was done using an homemade python script and has generated a new file that we called .GTF. This file was used to map the reads to the corresponding genomes using minimap2[88]. The .GTF and .sam files were used to perform the reads counts using featureCounts v2.0.1 of Subread package[89]. The count files for each sample were joined into a table using a homemade script and these results were analyzed using DESeq2 version 3.14[90]. Parameter used were Reads >1 in the 10 genomes (core-transcriptome: genes that were showing at least one read mapped in all genomes). We investigated the biological functions of the gene differentially expressed and the putative pathways that could link them through a STRING analysis[91].

## Quantitative real-time PCR

RNA samples were standardized to a final concentration of 1 μg with addition of DNaseI Amplification grade (Invitrogen) for genomic DNA removal. Random primers (Invitrogen), and RevertAid H-Minus reverse transcriptase (Thermo Scientific) were used for complementary DNA synthesis (cDNA) according to the manufacturer's instructions. Absence of contaminating gDNA was verified by conventional PCR of RNA samples in the absence of reverse transcriptase. Gene expression of *dcw* cluster was verified by quantitative real-time PCR (qRT-PCR) using Power SYBR Green PCR master mix (Applied Biosystems) using primers listed in Supplementary Table 3. Differential gene expression was calculated using ΔΔCT method using the mean CT value of each target obtained with the StepOne™ Software v2.3, normalization was done relative to *gyrA* gene. Standard T-test using (GraphPad Prism v9.0; GraphPad Software, CA) was used to ascertain statistical significance of gene expression between the strains, where $P < 0.05$ was considered significant.

## Genomic organization of the *dcw* cluster and *cdsA* loci in the *Neisseriaceae*

Coordinates of the *dcw* cluster and of the *cdsA* loci were obtained by tblastn for each *Neisseriaceae* genome. Once the genomic location of each sequence was determined, the sequences were extracted using tools available in the EMBOSS package[92]. The resulting sequences were annotated with Prokka, and the output gbk files were used to construct the synteny by employing Easy Fig 2.2.2[93].

## Construction of mutant strains

*Neisseria elongata* mutant strains were done in *N. elongata* subsp. *gly-colytica* (ATCC 29315) for single gene mutation and its streptomycin-resistant variant with a point mutations K88R *rpsL*\* for unmarked and multiple gene editing. *mraZ* was deleted by replacing *mraZ* with an mCherry-encoding gene. The construct for *mraZ* deletion was obtained by fusing multiple PCR fragments using Phusion DNA polymerase according to the protocol (New England Biolabs) as follows: firstly, *N. elongata* gDNA was used to amplify ~500 bp of regions up and down stream of *mraZ* using, respectively, primer pairs 5′KoMraZF-R and 3′KoMraZF-R. The promoter "pdcwSm", located upstream the *S. muelleri dcw* cluster, was amplified from *S. muelleri* gDNA using primer pairs pdcwsmF/pdcwsmR. Primer pairs 5MraZKmF and KmpSimR were used to amplify the kanamycin resistance cassette from pGEM::Km plasmid DNA[16], while the Mcherry cassette was obtained by PCR amplification of pMcherry10 (Addgene) using primer pairs pdcwsmMcherry F and McherryNsilR. Subsequently, the 5′MraZ and Km cassette were fused using primer pairs 5KoMraZF and KmpSimR, while Mcherry and 3′MraZ were fused using primer pairs pdcwMcherryF and 3′KoMraZ R. Finally, 5′MraZ-KM, pdcwSm and Mcherry-3′MraZ fragments were fused using primer pairs 5KoMraZ F and 3KoMraZ R and the resulting DNA was used for transformation in *N. elongata*.

To overexpress *mraZ*, *Neisseria meningitidis* promoter, *porB* was amplified from *N. meningitis* gDNA using primer pairs (porBpF-porBpbluntR) while the *mraZ* gene was amplified from *N. elongata* gDNA using primer pairs (MraZSphIF-3MraZR). The *porB* promoter from *N. meningitidis* and the *mraZ* gene from *N. elongata* were subsequently fused by PCR. This resulted in an ~1.6 kb-long porBp*mraZ* cassette that was digested using the restriction enzymes NheI and KpnI and then ligated with NheI-KpnI digested plasmid p5nrq3::Cm[16]. The ligation mix was transformed in *E. coli* DH5α cells to obtain the porBMraZ::p5nrq3::Cm plasmid. The plasmid was subsequently linearized before transformation into the *Neisseria elongata ΔmraZ* strain.

For the single knockout of *ΔmraZ*, *ΔrapZ*, *ΔgloB* or *Δdgt*, we used a cassette developed in our laboratory named RPLK[15] that contains the wild-type *N. elongata rpsL* gene, *N. meningitidis* promoter *porBp*, the blue-white screening selection marker *lacZ* and the kanamycin resistance marker that facilitated the generation of unmarked deletion in addition to multiple gene editing. We used synthesized DNAs (Bio-Basic) that contain ~500 bp each 5′ and 3′ regions surrounding the respective genes with a central BglII restriction site and cloned into pUC57 plasmid. The plasmids were linearized using BglII and ligated with RPLK cassette[15]. Mutants were obtained by transforming either *N. elongata* wild-type (single knockout) or an *N. elongata* streptomycin-resistant strain (indicated *rpsL*\*) (multiple knockout) with the linearized plasmid of the targeted gene that resulted in blue, kanamycin-resistant, streptomycin sensitive clones. Markerless deletion was achieved by introducing DNA of the 5′−3′ homologous regions of the target gene thereby excising the RPLK cassette resulting in white, kanamycin sensitive and streptomycin resistant clones. Subsequent genes of interest were edited by repeating this procedure and ver-ifications of the correct excision was done by PCR.

For allelic switching of *N. elongata mreB* with that from *S. muelleri*, plasmid pMreBSimon-3′RD3Ne was obtained by amplifying *S. muelleri mreB* using primer pairs MreBsimonF − MreBsimonR, while the sub-sequent region of the locus (3′RD3Ne that comprise a piece of *mreCD*)

was amplified from *N. elongata* using primer pairs 3′RD3NeF-3′RD3NeR. The two products were fused using primer pairs MreBsi-monF − 3′RD3NeR. This generated a cassette of *mreB*$_{sm}$ fused with *mreCD*$_{ne}$ that was then digested by restriction enzymes BamHI and SpeI before ligation with plasmid p5KORD1Ne::cm[16] digested with the same enzymes to obtain plasmid pMreBsimon3′RD3Ne::cm. The plas-mid was linearized with ScaI before transformation in *N. elongata* strains. *mreB*$_{sm}$ positive and *mreB*$_{ne}$ negative clones were con-firmed by PCR.

For the *cdsA-amiC2* knock-in constructs, we used the plasmid pUCNe::ampR that contains 5′ and 3′ *Neisseria elongata* homologous regions to the intergenic locus between two genes coding for hypo-thetical proteins at position 888015 (insertion site). We first con-structed the pUCNe::RPLK plasmid by ligating the RPLK cassette using BglII. Secondly, *cdsA-AmiC2* PCR product was obtained using primer pairs cdsAmiC2F-amiC2R, was digested using BglII and ligated to pUCNe::ampR to produce the pUCcdsamiC2::ampR plasmid. The mutants were obtained with a two-step method[15]. First, we trans-formed the plasmid pUCNe::RPLK into *N. elongata rpsL*\* to obtain *N. elongata* RPLK (RPLK inserted at position 888015). In the second step, we have replaced the RPLK cassette with *cdsA-amiC2* genes, by trans-forming the pUCcdsamiC2::ampR plasmid linearized using ScaI into *N. elongata* RPLK. *cdsamic2* positive transformants were con-firmed by PCR.

## Reporting summary

Further information on research design is available in the Nature Research Reporting Summary linked to this article.

## Data availability

The genome datasets generated during and/or analyzed during the current study (see Supplementary Data 1) are available in the NCBI genome repository (https://www.ncbi.nlm.nih.gov/genome/browse#!/overview/) under the accession codes: GCA_022870985.1, GCA_014055025.1, GCA_000818035.1, GCA_022870825.1, GCA_022870885.1, GCA_900637855.1, GCA_008807015.1, GCA_014055005.1, GCA_014297595.1, GCA_001308015.1, GCA_014054885.1, GCA_900636765.1, GCA_900638685.1, GCA_022870865.1, GCA_022870845.1, GCA_022870905.1, GCA_002951835.1, GCA_014054525.1, GCA_022871045.1, GCA_900177895.1, GCA_022871005.1, GCA_014054985.1, GCA_016623605.1, GCA_016127355.1, GCA_014054725.1, GCA_022871025.1, GCA_000745895.1, GCA_022870965.1, GCA_022870945.1, GCA_022870925.1, GCA_001648355.1, GCA_001648475.1, GCA_008805035.1, GCA_900187105.1, GCA_014054965.1, Raw reads data are available on SRA database under the acces-sion codes: PRJNA788950, PRJNA859696, PRJNA859916, PRJNA859935. Source data and the corresponding statistics are provided as a Source Data file and at the Cell Image Library repository [https://doi.org/10.7295/W9NC5ZC0]. Source data are provided with this paper.

## Code availability

The codes used in this study have been reported previously and are available as described in the corresponding M&M section. The doc-umentation for the ImageJ plugin Fil-Tracer can be accessed here: https://sils.fnwi.uva.nl/bcb/objectj/examples/Fil-Tracer/MD/Fil-Tracer.html. The other custom codes generated during the current study are available from the corresponding authors on reasonable request.

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

## Acknowledgements

The authors are extremely grateful to: Markus Christian Schmid and the Department of Microbial and Ecosystem Science of the University of Vienna for providing the Leica SP8 confocal laser scanning microscope and superb technical support; Belma Bejtovic and Mary Ward for performing preliminary experiments; Nathan Weyand for providing us with *Neisseria* sp. DentCa1/247; Tanneke den Blaauwen (University of Amsterdam) for inspiring and constructive discussions; Antony Vincent (University Laval) for his help in genome assemblies; Dennis Claessen for valuable comments on the manuscript. This work was supported by the Natural Sciences and Engineering Research Council of Canada (NSERC) discovery grant (RGPIN-2016-04940) (F.V.), the Fonds De Recherche du Quebec Nature et technologies (FRQNT) Établissement de la relève professorale (205027) (F.V.) and the Austrian Science Fund (FWF) project P28593-B22 (S.B., P.M.W., T.V., N.K.). P.M.W. also received DOC-fellowship 25240 from the Austrian Academy of Science and a PhD completion grant from the University of Vienna. E.B. received a Ph.D. Fellowship from the Fondation Armand-Frappier. M.D. was partially supported by a postdoctoral fellowship from the Swiss National Science Foundation (project #P2GEP3_191489). C.N. Received a Ph.D. studentship Calmette & Yersin from the Institut Pasteur International Network. Research in the Cava lab was supported by The Swedish Research Council (VR), The Knut and Alice Wallenberg Foundation (KAW), The Laboratory of Molecular Infection Medicine Sweden (MIMS), and The Kempe Foundation. M.N. was funded by a postdoctoral fellowship from the Swedish Society for Medical Research (SSMF). Y.V.B. is also supported by a Canada 150 Research Chair in Bacterial Cell Biology. F.J.V. received a Junior 1 and Junior 2 research scholar salary award from the Fonds de Recherche du Québec - Santé. The funders had no role in study design, data collection and analysis, decision to publish, or preparation of the manuscript.

## Author contributions

S.N. and P.M.W did most experiments, visualization and formal analysis, wrote and revised the manuscript. E.B.; F.P; M.N.; M.D; C.N.; T.V.; N.K.; A.R.M. and A.N. did some experiments and formal analysis and critically revised the manuscript. N.O.E.V. contributed ImageJ analysis tools (ObjectJ, Fil-Tracer). M.V. contributed materials. Y.B. acquired funding and analysis tools. F.C. acquired funding, did formal analysis and revised manuscript. S.B. conceptualized and supervised the work, acquired funding, provided resources, wrote and revised the manuscript. F.J.V. did experiments, formal analysis, conceptualized and supervised the work, acquired funding, provided resources, wrote and revised the manuscript. Equally contributing authors were listed in alphabetical order.

## Competing interests

The authors declare no competing interests.

## Additional information

**Supplementary information** The online version contains

supplementary material available at https://doi.org/10.1038/s41467-022-32260-w.

