## [Peer Review File · Nature Communications]

Reviewers' Comments:

Reviewer #1:

Remarks to the Author:

The manuscript by Nyongesa and colleagues describes an in-depth investigation of the evolution and molecular mechanisms of the evolution of multicellular longitudinal division in the Neisseriaceae. The work is extremely well conceived, executed, and presented. There is a wonderful mix of evolutionary work, electron and fluorescence microscopy, cross-species differential transcriptomics, and analysis of mutants that culminates in solid descriptions of multicellularity in *Alysiella*, *Simonsiella*, and *Conchiformibius*. I do have a few suggestions for improvement, as summarized below. My hope is that these suggestions lead to an improved manuscript.

Scientific issues:

1. line 181 suggests "multicellularity results from adjoining cells attached by a cell-wall (should have no hyphen here) flanked by two inner membranes. I can't make sense of the interpretation here. I think these organisms have a typical Gram-negative envelope structure with peptidoglycan flanked by the cytoplasmic membrane and outer membrane (not two inner membranes). Is that what is being described? If so, please improve language. I suggest using "peptidoglycan" or "peptidoglycan layer" as a clearer communication than "cell wall".

If my interpretation above is correct, is the suggestion that a filament would be surrounded by a shared outer membrane and periplasm similar to *Dictyoglomus*? If not, I can't understand how the cells would be attached by the peptidoglycan (as is clearly the case based on the sacculi), unless somehow the peptidoglycan pierces the outer membrane at the cell junctions, which doesn't make sense. Please consider this carefully and improving the interpretation, description, and possibly figures (or supplemental figures) so the logic is clear.

2. line 324 and elsewhere: the *mraZ* deletion strain appears to have a larger cell volume (possibly due to larger diameter), but only cell length is plotted. Is there a reason diameter or modeled volume aren't compared? I realize that modeled volume would have a propagation of error, but it would also magnify differences due to differences in two dimensions and still be biologically important. Anyway, something to consider. Please don't change this if you don't feel it's an improvement.

3. The phylogenetic analyses (and resulting interpretations) are not as clearly presented as other parts of the paper, which limits the clarity of the interpretation of the evolution of multicellularity and longitudinal cell division. Notably, the results section "Core genome-based phylogeny of Neisseriaceae suggests that MuLDi evolved from rod-shaped ancestor", does not sufficiently describe the results to support this interpretation, and certainly not the interpretation that "two lineages of cocci...evolved independently from a rod-shaped ancestor and that two lineages of MuLDi evolved from a rod-shaped ancestor" (162-165). This latter text is contradicted by later text (line 382) that MuLDi may have evolved either once or twice. Actually, this section only has two sentences describing the phylogeny in this results section: "We obtained 69 species for the construction of a core genome-based phylogeny using closed genomes or, if not, the draft genome for each species (Figure 1). Phylogeny results were similar to a recently published study (Chen et al., 2021)."

Some specific recommendations and detailed criticisms:

- I think it would be good to list more about the phylogeny in the results section, for example a description of how many core genes were used and better description of the structure of the tree.
- Very importantly, although the text says that the phylogeny is similar to that in Chen et al., 2021, there are key differences that influence the interpretation of the evolution of MuLDi. In Chen et al., 2021, the genus *Kingella* is monophyletic within the MuLDi group. If this is true, then most likely MuLDi evolved at a single node and then *Kingella* reverted to rod-shaped morphology. On the contrary, the phylogeny reported in the current paper is more difficult to interpret because *Kingella* is not monophyletic, requiring a more complex evolutionary scenario. This difference needs to be resolved or, if not, discussed.
- Although 10,000 bootstraps were done on the core gene phylogeny, bootstrap support is not

included in the tree, so the support for the key nodes discussed above are not shared. Similarly, no bootstrap values are provided in the AmiC phylogenies and since the AmiC trees are so large there's no way to evaluate whether the species tree (Figure 1) matches the AmiC phylogenies, which would help to interpret the evolution of multicellularity. More detail of the AmiC trees is needed to see whether 1) AmiC1 reflects the species tree topology (which superficially looks to be true, but no way of actually evaluating that truly), and 2) whether the AmiC2 lineage also follows the evolutionary history of the species tree (i.e., if *Simonsiella* is sister to *Alysiella*, followed by *Conchiformibius*), then this likely supports vertical inheritance because it reflects the species tree topology, and a single loss in *Kingella* occurred (a genus that is recovered as monophyletic elsewhere), but if any of these are nested within any of the other lineages, then it was likely an HGT event from one MuLDi lineage to another. So, we need specifics to make sense of this. Additionally, it would be interesting to know whether *Fusobacterium* is also the closest hits to the other MuLDi-specific genes (just a BLAST or something), but not required.

- As an additional suggestion, I would like to see a tree of the MreB and FtsA sequences too (with reference included), to see whether those amino acid changes were introduced once in the ancestor to the MuLDi lineage, and selected against in *Kingella*, or whether convergent evolution occurred in the two distinct lineages. Because *Simonsiella* and *Conchiformibius* share a phenotype, and *Alysiella* differs in how its multicellular filaments present, could lend credence to the HGT hypothesis from *Simonsiella* to *Conchiformibius* perhaps. Anyway, the evolutionary picture of these might be really interesting, and I feel like the paper missed out on something potentially awesome.

- Once the issues above are reconciled, please make sure the text in line 162 and line 382 present a coherent message.

Minor editorial comments:

- line 38 suggest "applying a palette of fluorescent peptidoglycan..."
- line 51: remove comma
- line 70 hard stop after "vertebrates"
- Line 71: besides being multicellular...
- line 125: "cultivation"
- line 132 and throughout: Suggest writing the rank before taxonomic name "family Neisseriaceae"
- line 144 – 147: Sentence is long and complex. Please reformulate.
- line 175: that rather than which
- line 178 sentence is complex and too many commas interrupting. Consider: "We observed that the sacculi of the three MuLDi symbionts remained attached laterally, even after the harsh extraction procedure. (or you could remove my extra words.)"
- line 195 and elsewhere, please correct problems with in-text references listing first names.
- line 199, "when observed by fluorescence microscopy"
- line 202, proceeded
- Figure 2 I,J,K, please label the taxa on the graphs/subgraphs for additional clarity.
- line 228, the antecedent for "they" is unclear because the previous paragraph is about fimbriae but this paragraph transitions to cell growth. Please specify that "they" is cells.
- line 254, detect
- line 256-259, if possible, please describe approach better here and specify that these genes are exclusively found in the two groups
- line 262, "included" rather than "comprised"
- line 264, "associated with" is ambiguous (i.e., chromosomally co-located or co-occurrence in genomes). Please clarify in text.
- Figure 4 please use a better term than "insertions" and "deletions" for the heatmap because you don't precisely describe the evolution of all of these genes. Consider "present" and "absent".
- Figure 5a, for a slightly easier understanding, consider changing the in-figure legend to "Genes downregulated in *N. elongate* del *mraZ* compared to [the complemented strain]". This is slightly easier because the text would be parallel in the figure and line 319 where it is described in the text (rather than the reverse).
- line 340, remove commas around "concomitantly"
- line 344, comma before "which" (almost always)
- line 355, replace "how many of them" with "the number that have been"

- line 389, remove comma after "although"
- line 401 "the cyanobacteria"
- line 408 (and abstract actually) need to write out each genus/species binomial before using acronyms.
- line 437 "that is conserved".
- line 454, which bacteria are filamented in MraZ overexpression strains? Please clarify
- Figure 1 legend, I can't distinguish two shades of blue

Reviewed by Brian Hedlund

Reviewer #2:

Remarks to the Author:

Nyongesa et al. describe the shape and peptidoglycan growth characteristics of three members of the Neisseriaceae, focusing on two species exhibiting an unusual multicellular morphology. Fluorescent labeling of peptidoglycan was used to examine the odd longitudinal septation phenotype. Genomic and transcriptomic methods were used to identify genes predicted to produce the multicellular phenotype. Mutants of *Neisseria elongata* were made in order to change the shape to mimic the multicellular species. The mutant bacteria were more similar to the multicellular species, but the shape change was not achieved.

Major points.

1. This is a straight-forward and mostly well-described study.
2. Support should be provided for the idea that *Alysiella*, *Simonsiella*, and *Conchifomibius* are multicellular. Is it being claimed that the bacterial groups have differentiated cells that perform different functions? If not, then aren't many bacterial species multicellular, e.g., streptococci growing in chains?
3. The figures need improvements. Increase the magnification for the TEMs of sacculi in Fig. 2 so that it is possible for the reader to judge if the sacculi contain intact walls between cells. Fig 2e. It does not appear that the boxed region corresponds to any of the cells shown in the high magnification photo. Fig. 3. Add species names to the panels. Fig. 4. Add more gene names to fig 4c, maybe five of them. Fig. 5. Explain the meaning of pilEp-mraZ. Is this an expression construct? Fig. 6 and elsewhere, correct rpsI to rpsL.
4. The authors should endeavor to make the results in Fig 2 and Fig 3 as easy as possible for the reader to understand. Walk the reader through these data.
5. Is the growth rate of the pilEp-mraZ strain different from wild type? Is it possible that the bacteria are short due to a growth abnormality?
6. AmiA, AmiB, and AmiC in *E. coli* use activators including NlpD, EnvC, and ActS (doi: 10.1111/mmi.14711). *N. gonorrhoeae* similarly uses NlpD to activate AmiC and has an envC gene, though its mutant has no morphology phenotype. It seems possible that AmiC2 described here also needs an activator.

Minor points.

1. Lines 152-155. This statement is not true for coccal *Neisseria* such as *N. gonorrhoeae*, *N. meningitidis*, *N. mucosa*, etc.
2. Line 318 and elsewhere, make sure that *N. elongata* is correctly spelled.
3. Line 320. It is not necessary to "validate" RNA-Seq results with real-time RT-PCR.
4. Line 376. This statement is too strong. Given the results presented, it is not necessarily true that the genes identified "contributed to the evolution of bacterial multicellularity". Reword to add a qualifier, such as "likely contributed" or "may have contributed".
5. Line 401. This sentence is awkward. Either delete "to" or change "resemble" to "similar".
Line 820. Indicate the source of fluorescent D-alanine analogs.
6. Fig. 1. Add a label for the blue cocci or define in the legend.
7. Fig. 6b is not mentioned in the manuscript.
8. Fig. S7 needs more labels and explanation.

Reviewer #3:

Remarks to the Author:

This manuscript provides landmark advances in our understanding the evolution of multicellular longitudinal cell division (MuLDi) in Neisseriaceae. By combining genome sequencing, microscopy and genetics they propose how this peculiar mode-of-division originated from an ancestral rod-shaped bacterium. Previous work from this lab pioneered this fascinating mode of division, and here they build upon that previous work by focusing on oral cavity symbionts belonging to the Neisseriaceae. The microscopy in this paper is outstanding, and the finding that this mode-of-division originated from a rod-shaped ancestor is clearly demonstrated by the phylogenetic reconstruction. The authors towards the end of the paper pioneer the first steps in reconstructing this division mode in the rod-shaped *Neisseria elongate*, and by introducing key changes based on their thorough phylogenetic analysis they manage to increase the size of septa (which in MuLDi strains is larger).

For their consideration, I have some suggestions that could improve this otherwise fantastic manuscript

1. I find the introduction, and in particular the first part (lines 56-86) quite intense in terms of the described organisms and their classification. There is a lot of information here on their morphology, classification, uni- versus multicellular etc (admittedly, I am not a phylogeny expert). Also, lines 83-86 quite abruptly start discussing fimbriae, which I understand later in the results, is important to determine the polarity. My advice would be to take out the fimbriae part in the introduction, and in the results section (lines 194-201) use one or two sentences that it is important to determine the polarity, for which antibodies against fimbriae are used.
2. Lines 169-174. I find this part oddly presented. I would start with the data (Fig. 1, 2, Fig. S1b-d, Sup Movies 1-4), and then compare to previous studies. Also, some text could be added to describe these results without just pointing to the Figures/Movies
3. In the parts describing how septa are formed in *A. filiformis*, *S. muelleri* and *C. steedae* I find myself switching between figures. Isn't it handier to organize the figures by species rather than imaging technique, in particular for the fluorescence panels? (lines 192-246)
4. The parts on (the absence of) *mraZ* could be combined into one paragraph (so the RNAseq and the mutational analysis in *N. elongate*).
5. In the discussion the authors speculate on the transition from unicellular to multicellular longitudinal division. What I miss in the discussion is something related to the organization and segregation of DNA. This should have also evolved, or is this passively following morphological changes? And could this be something that (partly) explains why the authors have not managed to reconstruct this mode-of-division?

Minor comments

Line 102: Ssg should be SsgB

Line 122: "...the acquisition OF *amiC2*..."

Line 139: "from THE NCBI databse..."

Line 154: light blue branches. To me all blue branches appear to have the same color.

Line 272: ShIB. Add reference to this claim.

Line 280. Why is *dgt/GloB* not discussed/mentioned?

Line 291: perhaps best to say that there is a correlation in the mode of division and the presence/absence of certain genes

Line 311-312. I think again this is correlation (so absence of *mraZ* correlates with different expression).

Line 334: data not shown: I think it's common to include this in Suppl. Information nowadays

Reviewer #4:

Remarks to the Author:

In this article entitled "Evolution of multicellular longitudinally dividing oral cavity symbionts (Neisseriaceae)", Nyongesa et al. present a model of cell division of the multicellular longitudinally dividing (MuLDi) organisms in Neisseriaceae and propose an evolutionary scenario of the emergence of such phenotype. First, the authors present a phylogenomic analysis that describe the diversity in term of shape and way to divide in Neisseriaceae and conclude that MuLDi clades emerged likely from a rod-shape ancestor. Then, they characterize the structure of the peptidoglycan using microscopy and mass spectrometry. They also describe the dynamic of muropeptide insertion into the mid-cell and conclude that the separation of the daughter cells is performed by PG sheet insertion, asynchronously for *A. filiformis* and synchronously for *S. muelleri* and *C. steedae*. Next, they identified the key evolutionary events that occurred at the stems of both MuLDi clades, involving notably major cell division and cell wall synthesis genes. Finally, the authors study deeper the effect of such evolutionary events by generating mutants of *mraZ* and other cell division genes that were gained, lost or mutated.

The known diversity of prokaryotes increased dramatically in the last decades, although most of the cell processes (and cell division) are still studied in a very few model organisms (rod-shape for most of them). In this context, the study of cell division mechanisms in non-model and non-rod-shaped bacteria is of particular interest. The authors combined elegantly phylogenomics, cell biology and biochemistry to decipher both mechanism and evolution of the cell division of the MuLDi. The results are interesting and convincing and the text is globally well-written.

Nevertheless, as a specialist of phylogenomics, I have a few recommendations to the authors to strengthen the genomic analysis and to better clarify some paragraphs.

Methodological comments:

The methods for the genomic analysis are not very clear (l. 131-147 and 943-979), there are some discrepancies between supplementary tables and the text, and some information/data are missing. To clarify this:

- Write an entire section of M&M with the detail of the method of the reduction of the number of genomes from 21+365 genomes to 69 genomes (Download of NCBI genomes, ANI, clustering method, selection of the best genome per cluster of species).
- To make it easier for non-specialists, explain more clearly the rationale for each step, in both result and M&M sections (Why sequencing these 21 genomes, why mixing them with NCBI genomes, why comparing genomes with ANI, why reducing the number of genomes, ...).
- Indicate the software that has been used for ANI and the parameters in M&M. Also, give the parameters used for each software (Roary, MycoHIT, CapriB, ..).
- Provide the raw data of phylogenetic analysis (Alignments, Supermatrix, Tree with bootstrap values). Give the list of markers (core genome) in a table, the total number of markers and the number of positions of the alignment that have been used to build the phylogeny of Neisseriaceae. Also, provide the exact evolutionary model that has been selected by IQ-TREE.
- Indicate clearly the source (collection like DSMZ or samples?) of the newly sequenced genomes, in both result and M&M sections.
- Highlight clearly the 21 genomes in the table S2.
- Table S1 and S2 seem to be inverted in the text (l. 137 and 140).
- L. 109: I see only 41 genomes in table S2, not 42.
- I suppose that the 69-41 = 28 genomes correspond to the cocci clade 1. List these genomes in table S2 and indicate clearly that 69-41 corresponds to the number of genomes inside this clade in the text.
- Table S1: there are only 261 genomes instead of 365+21 genomes. Complete the matrix (even if ANI <70%). Also, provide a list of the 365 genomes originating from NCBI with proper identifiers.

The authors did not use a statistical method to infer the ancestral states (rod-shaped/cocci/MuLDi) (l. 158-161). The inference of the ancestral states using a Maximum Likelihood method such as PastML could reinforce the data. It is an easy and quick analysis and I have no doubt that the results of such method would be in agreement with the manual inferences presented in Figure 1. The only potential issue would be the 8 lineages for which there is no microscopy data, but they could be removed from the tree to perform the analysis. Also, the authors could clearly indicate each genome for which microscopy data are available in table S2.

In any case, the argument of the wide distribution of rod-shaped species is not sufficient to infer the rod-shape nature of the ancestor. The rod-shape nature of basal lineages (outgroup, E.

corrodens clade, V. massiliensis clade, ...) is crucial to such conclusion and this should be stated in the text.

Concerning the pangenome analysis (l. 958-959 in M&M), the use of the set of proteins of two reference genomes can be hazardous as some proteins could be only absent in the reference genomes but still widely present in the other genomes. The use of a simple method without a priori like whole database clustering methods (MMseqs2, SiLiX, HiFiX, Roary) would be more suitable for such analysis.

Other comments:

l. 152-155: This sentence and especially "Neisseriaceae are rod-shaped, except for two closely related species" is misleading as the figure and the text l. 163-165 highlight not only this clade as an exception, but two cocci clades and two MuLDi clades. Please rephrase.

l. 155-156: Please explain very shortly the method that has been used to infer the loss of such genes in the text.

Figure 1: I have a few comments:

- The difference between light and dark blue is not visible.
- One microscopy image is not connected to any tip.
- Use 1 and 2 for both cocci and MuLDi clades is confusing, maybe 1/2 and A/B?

l. 180-181: This model of division is really intriguing and rises some questions. Does it mean that cells are encapsulated by the outer membrane like a "plastic film of a six-pack of bottles", or is it possible that the outer membrane is still present in the mid-cell? The first hypothesis would mean that another secondary division occurs (when the cells are really separated during the chain growth) to lyse the peptidoglycan and invaginate the outer membrane. In other model bacteria, the invagination of the outer membrane is driven by septation. So, what could be the driving force for such a case? I think that such hypotheses/questions should be developed in the discussion section.

l. 210: "with what "is" observed"?

l. 264 and 469: Choose either phosphatidate cytidyltransferase or CDP-diacylglycerol synthase to describe CdsA function.

l. 445: For your information, it has been suggested that the presence/absence of MreB and Min genes is responsible for the rod-shape/cocci phenotype, and not particularly the fragmentation of the DCW cluster (How to Build a Bacterial Cell: MreB as the Foreman of E. coli Construction, Shi et al. 2018, A Comprehensive Evolutionary Scenario of Cell Division and Associated Processes in the Firmicutes, Garcia et al., 2020).

l. 478: Another key event that could have occurred is the rearrangement of the order of genes within the genome. It could be interesting to explore this point (But I do not ask the authors to do it).

Point-to-point rebuttal

Reviewer #1 (Remarks to the Author):

The manuscript by Nyongesa and colleagues describes an in-depth investigation of the evolution and molecular mechanisms of the evolution of multicellular longitudinally division in the Neisseriaceae. The work is extremely well conceived, executed, and presented. There is a wonderful mix of evolutionary work, electron and fluorescence microscopy, cross-species differential transcriptomics, and analysis of mutants that culminates in solid descriptions of multicellularity in *Alysiella*, *Simonsiella*, and *Conchiformibius*. I do have a few suggestions for improvement, as summarized below. My hope is that these suggestions lead to an improved manuscript.

Scientific

issues:

1. line 181 suggests “multicellularity results from adjoining cells attached by a cell-wall (should have no hyphen here) flanked by two inner membranes. I can’t make sense of the interpretation here. I think these organisms have a typical Gram-negative envelope structure with peptidoglycan flanked by the cytoplasmic membrane and outer membrane (not two inner membranes). Is that what is being described? If so, please improve language.

Thanks a lot for this comment. We have added higher magnification TEM images (see revised Figure 2) showing that each MuLDi *Neisseriaceae* cell has its own inner membrane (IM) but shares its lateral PG. Furthermore, cells belonging to the same filament share their outer membrane (OM) and, likely, their periplasm.

Concerning how this “permanent chaining” is achieved, based on the available ultrastructural analysis (see, for example, *S. muelleri* filament in the Figure below, which we have included as Figure S3 of the revised manuscript), we hypothesize two subsequent rounds: during the first one, the MuLDi synthesize a septum composed of one sheet of PG (see nascent and completed septa below). This PG sheet is flanked by an IM at each side. During the second round (septum between cell 2 and cell 3 below) and starting at the poles, the septum appears to split into two PG sheets, concomitantly with the OM invagination until midcell is reached. Because the cells share the OM, Reviewer #4 compared the cells to bottles in a six-pack (and the OM would therefore correspond to the plastic film wrapping them). To clarify that the OM is shared by cells belonging to the same filament, we have improved Figure 2 and we have added the Figure below as Figure S3.

If my interpretation above is correct, is the suggestion that a filament would be surrounded by a shared outer membrane and periplasm similar to Dictyoglomus?

Correct. We have clarified this in the revised Main text and Figures 2 and S3.

If not, I can't understand how the cells would be attached by the peptidoglycan (as is clearly the case based on the sacculi), unless somehow the peptidoglycan pierces the outer membrane at the cell junctions, which doesn't make sense. Please consider this carefully and improving the interpretation, description, and possibly figures (or supplemental figures) so the logic is clear.

Please read rebuttal to point 1.

I suggest using "peptidoglycan" or "peptidoglycan layer" as a clearer communication than "cell wall".

We agreed and used "peptidoglycan" instead of "cell wall" throughout the manuscript.

2. line 324 and elsewhere: the *mraZ* deletion strain appears to have a larger cell volume

(possibly due to larger diameter), but only cell length is plotted. Is there a reason diameter or modeled volume aren't compared? I realize that modeled volume would have a propagation of error, but it would also magnify differences due to differences in two dimensions and still be biologically important. Anyway, something to consider. Please don't change this if you don't feel it's an improvement.

Thanks a lot for this comment. We did the suggested measurements, but we did not observe an increase in diameter (Figure A, below). Therefore, the volume differences observed (Figure B, below) are due to cell length differences observed in Figure 6. As we did not see differences in the diameter, we prefer not to include this, unless the Reviewer wishes so.

3. The phylogenetic analyses (and resulting interpretations) are not as clearly presented as other parts of the paper, which limits the clarity of the interpretation of the evolution of multicellularity and longitudinal cell division. Notably, the results section "Core genome-based phylogeny of Neisseriaceae suggests that MuLDi evolved from rod-shaped ancestor", does not sufficiently describe the results to support this interpretation, and certainly not the interpretation that "two lineages of cocci...evolved independently from a rod-shaped ancestor and that two lineages of MuLDi evolved from a rod-shaped ancestor" (162-165). This latter text is contradicted by later text (line 382) that MuLDi may have evolved either once or twice. Actually, this section only has two sentences describing the phylogeny in this results section: "We obtained 69 species for the construction of a core genome-based phylogeny using closed genomes or, if not, the draft genome for each species (Figure 1). Phylogeny results were similar to a recently published study (Chen et al., 2021)." Some specific recommendations and detailed criticisms:

- I think it would be good to list more about the phylogeny in the results section, for example a description of how many core genes were used and better description of the structure of the tree.

We agree and have extended the Methods section accordingly. We originally kept it concise because our phylogenetic analysis was performed similarly to that published by Chen et al., 2021, except that we used more stringent criteria to select the genes we used for the phylogenetic reconstruction. Namely, we used 55% of identity and presence of the genes in 95% of the strains, whereas Chen et al., 2021 used 50% of identity and presence of the genes in 80% of the genomes. These differences led to different phylogenetic trees. In the revised manuscript, to avoid confusion, we have applied the same parameters applied by Chen et al., 2021. This led to the clustering of all the *Kingella* spp. in one lineage in agreement with the previous work by Chen et al., 2021. This revised phylogeny is now presented in the revised Figure 1 and we also provide the number and identity of genes we used to obtain it (Table S3).

Finally, following the suggestion of Reviewer 4, we have provided the PastML analysis to infer the ancestral shape of the *Neisseriaceae* (see Figure S1 of the revised manuscript) and acknowledged that the phylogeny at the node *Kingella*M1/M2 may not be sufficiently resolved. We are planning on isolating more *Kingella* and MuLDi species to improve the resolution.

- Very importantly, although the text says that the phylogeny is similar to that in Chen et al., 2021, there are key differences that influence the interpretation of the evolution of MuLDi. In Chen et al., 2021, the genus *Kingella* is monophyletic within the MuLDi group. If this is true, then most likely MuLDi evolved at a single node and then *Kingella* reverted to rod-

shaped morphology. On the contrary, the phylogeny reported in the current paper is more difficult to interpret because *Kingella* is not monophyletic, requiring a more complex evolutionary scenario. This difference needs to be resolved or, if not, discussed.

We agree with the Reviewer and revised accordingly (see rebuttal to the previous point). We now propose alternative evolutionary scenarios in both the revised Results and Discussion. Moreover, we clearly state that we currently cannot unambiguously exclude any of them. In addition to this, we have added new data for gene insertion/deletion that excluded the *Kingella* spp. from the analyses (Table S4) and that have been possibly acquired or lost in the most recent common ancestor of M1, M2 and *Kingella* spp.. Finally, in agreement with the Reviewer #4, we have used the PastML tool and confirmed that MuLDi and cocci evolved from a rod-shape ancestor.

- Although 10,000 bootstraps were done on the core gene phylogeny, bootstrap support is not included in the tree, so the support for the key nodes discussed above are not shared. We have now added the bootstrap values when they were lower than 100.

Similarly, no bootstrap values are provided in the AmiC phylogenies and since the AmiC trees are so large there's no way to evaluate whether the species tree (Figure 1) matches the AmiC phylogenies, which would help to interpret the evolution of multicellularity. More detail of the AmiC trees is needed to see whether 1) AmiC1 reflects the species tree topology (which superficially looks to be true, but no way of actually evaluating that truly), and 2) whether the AmiC2 lineage also follows the evolutionary history of the species tree (i.e., if *Simonsiella* is sister to *Alysiella*, followed by *Conchiformibius*), then this likely supports vertical inheritance because it reflects the species tree topology, and a single loss in *Kingella* occurred (a genus that is recovered as monophyletic elsewhere), but if any of these are nested within any of the other lineages, then it was likely an HGT event from one MuLDi lineage to another. So, we need specifics to make sense of this. Additionally, it would be interesting to know whether *Fusobacterium* is also the closest hits to the other MuLDi-specific genes (just a BLAST or something), but not required.

Thanks a lot for this comment. We have revised Figure S10 to include all the suggested phylogenetic analyses (main, AmiC1, AmiC2, MreB and FtsA), as well as those of RapZ and MraZ. Moreover, in the revised Main text, we added possible evolutionary scenarios.

- As an additional suggestion, I would like to see a tree of the MreB and FtsA sequences too (with reference included), to see whether those amino acid changes were introduced once in the ancestor to the MuLDi lineage, and selected against in *Kingella*, or whether convergent evolution occurred in the two distinct lineages. Because *Simonsiella* and *Conchiformibius* share a phenotype, and *Alysiella* differs in how its multicellular filaments present, could lend credence to the HGT hypothesis from *Simonsiella* to *Conchiformibius* perhaps. Anyway, the evolutionary picture of these might be really interesting, and I feel like the paper missed out on something potentially awesome.

We agree and we have now provided the phylogenetic trees for RapZ, MraZ, AmiC1, MreB and FtsA in Figure S10. AmiC1 is conserved in all species and its phylogeny mimics the genome-based phylogeny. RapZ and MraZ (absent in MuLDi) phylogenies agree with a vertical transmission of the genes encoding for these two proteins in the *Kingella* genus (and therefore independent loss in M1 and M2). Finally, phylogenies based on either MreB or FtsA (two proteins expressed by all *Neisseriaceae*, but bearing specific amino acid permutations in MuLDi *Neisseriaceae*) indicate that both MuLDi MreBs (or FtsAs) cluster together and separately from *Kingella* spp. These latter two phylogenies support convergent evolution of amino acid substitutions in MuLDi *Neisseriaceae* and hint at the importance of MreB and FtsA in the evolution of the MuLDi phenotype.

- Once the issues above are reconciled, please make sure the text in line 162 and line 382 present a coherent message.

The Main text is now coherent, thank you.

Minor editorial comments:

- line 38 suggest "applying a palette of fluorescent peptidoglycan..."

- line 51: remove comma

- line 70 hard stop after “vertebrates”
 - Line 71: besides being multicellular...
 - line 125: “cultivation”
 - line 132 and throughout: Suggest writing the rank before taxonomic name “family Neisseriaceae”
 - line 144 – 147: Sentence is long and complex. Please reformulate.
 - line 175: that rather than which
 - line 178 sentence is complex and too many commas interrupting. Consider: “We observed that the sacculi of the three MuLDi symbionts remained attached laterally, even after the harsh extraction procedure. (or you could remove my extra words.)
 - line 195 and elsewhere, please correct problems with in-text references listing first names.
 - line 199, “when observed by fluorescence microscopy”
 - line 202, proceeded
 - Figure 2 I,J,K, please label the taxa on the graphs/subgraphs for additional clarity.
 - line 228, the antecedent for “they” is unclear because the previous paragraph is about fimbriae but this paragraph transitions to cell growth. Please specify that “they” is cells.
 - line 254, detect
 - line 256-259, if possible, please describe approach better here and specify that these genes are exclusively found in the two groups
 - line 262, “included” rather than “comprised”
 - line 264, “associated with” is ambiguous (i.e., chromosomally co-located or co-occurrence in genomes). Please clarify in text.
 - Figure 4 please use a better term than “insertions” and “deletions” for the heatmap because you don’t precisely describe the evolution of all of these genes. Consider “present” and “absent”.
 - Figure 5a, for a slightly easier understanding, consider changing the in-figure legend to “Genes downregulated in *N. elongata* del *mraZ* compared to [the complemented strain]”. This is slightly easier because the text would be parallel in the figure and line 319 where it is described in the text (rather than the reverse).
 - line 340, remove commas around “concomitantly”
 - line 344, comma before “which” (almost always)
 - line 355, replace “how many of them” with “the number that have been”
 - line 389, remove comma after “although”
 - line 401 “the cyanobacteria”
 - line 408 (and abstract actually) need to write out each genus/species binomial before using acronyms.
 - line 437 “that is conserved”.
 - line 454, which bacteria are filamented in *MraZ* overexpression strains? Please clarify
 - Figure 1 legend, I can’t distinguish two shades of blue
- We agree and addressed all the minor editorial comments, thank you!

Reviewer #2 (Remarks to the Author):
 Nyongesa et al. describe the shape and peptidoglycan growth characteristics of three members of the Neisseriaceae, focusing on two species exhibiting an unusual multicellular morphology. Fluorescent labeling of peptidoglycan was used to examine the odd longitudinal septation phenotype. Genomic and transcriptomic methods were used to identify genes predicted to produce the multicellular phenotype. Mutants of *Neisseria elongata* were made in order to change the shape to mimic the multicellular species. The mutant bacteria were more similar to the multicellular species, but the shape change was not achieved.

Major points.

1. This is a straight-forward and mostly well-described study.
2. Support should be provided for the idea that *Alysiella*, *Simonsiella*, and *Conchifomibius* are multicellular. Is it being claimed that the bacterial groups have differentiated cells that perform different functions?

If not, then aren't many bacterial species multicellular, e.g., streptococci growing in chains?

At the moment we do not have any evidence of cell differentiation within MuLDi filaments. However, within filament-differentiation has been suggested for *Simonsiella muelleri* because the filament terminal cells are morphologically different from the central ones (Hedlund 2015) and perhaps this also occurs in *C. steedae* (see cells in white square insets in Figure below; P. Weber, unpublished). Moreover, in *C. steedae* we observed a thinner cell every 14 cells, approximately (Figure S8a of revised manuscript).

To conclude, we agree with the Reviewer that the term “multicellularity” traditionally entails the phenomenon of “cell differentiation”. However, even if within filament cell differentiation needs to be proven, we would still like to refer to the MuLDi *Neisseriaceae* as multicellular, because: (1) they share their lateral PG and (2) they invariably form chains of >2 cells. This is stated in the Introduction (i.e., what we mean by “multicellular” in this manuscript) and in the Discussion we state that we currently have no evidence of division of labour among cells belonging to the same filament.

3. The figures need improvements. Increase the magnification for the TEMs of sacculi in Fig. 2 so that it is possible for the reader to judge if the sacculi contain intact walls between cells.

The Figure was improved as suggested, thank you.

Fig 2e. It does not appear that the boxed region corresponds to any of the cells shown in the high magnification photo.

We corrected this.

Fig. 3. Add species names to the panels.

We corrected this.

Fig. 4. Add more gene names to fig 4c, maybe five of them.

We corrected this.

Fig. 5. Explain the meaning of pilEp-mraZ. Is this an expression construct?

We apologize for mistakenly have written *pilEp-mraZ* instead of *porBp-mraZ*. *porBp* (nomenclature suggested by ASM journals) refers to the promoter of the *porB* gene which we use to overexpress proteins in *Neisseriaceae* in this work. Although the promoter of the *pilE* gene (*pilEp*) may also be used to overexpress proteins in *Neisseriaceae*, we did not use this promoter in this work. We thank the Reviewer for making us notice the mistake and we have revised accordingly.

Fig. 6 and elsewhere, correct rpsI to rpsL.

It is now rpsL throughout the text and the figures.

4. The authors should endeavor to make the results in Fig 2 and Fig 3 as easy as possible for the reader to understand. Walk the reader through these data.

We modified the main text and the figures as suggested.

5. Is the growth rate of the pilEp-mraZ strain different from wild type? Is it possible that the bacteria are short due to a growth abnormality?

The grow rate is similar between the different strains (see Figure below), suggesting that the cell-shape changes are not impacting the growth significantly.

6. AmiA, AmiB, and AmiC in E. coli use activators including NlpD, EnvC, and ActS (doi: 10.1111/mmi.14711). N. gonorrhoeae similarly uses NlpD to activate AmiC and has an envC gene, though its mutant has no morphology phenotype. It seems possible that AmiC2 described here also needs an activator.

We agree and we plan to test whether the suggested proteins may activate AmiC2 in MuLDi Neisseriaceae. Of note, *nlpD* and *envC* are present in MuLDi but the conservation of their sequence is low. NELON_RS02280 (possibly encoding for ActS) is less conserved.

EnvC	nlpD	EnvC	nlpD
NELON_RS03310	NELON_RS09315	NELON_RS02280	NELON_RS02280
Neisseria dentica 1247			
Neisseria zoodiagmatis			
Neisseria aminobars			
Neisseria actica DSM10316			
Neisseria dentae			
Neisseria dumajana DSM10677			
Neisseria lazalogni			
Neisseria musculli			
Urubunella laus			
Urubunella stunsis DSM6650			
Neisseria sp. K122			
Neisseria laweveri			
Neisseria onis			
Neisseria dawsonii			
Neisseria sp. KEM232			
Neisseria papus			
Neisseria bacilliformis			
Neisseria elongata ATCC29315			
Simondella mellei 2943			
Alpella crassa			
Alpella filiformis			
Kingella negevensis			
Kingella onis			
Kingella bonacopii			
Kingella kingae			
Kingella dentrificans			
Conchiformis kulinae			
Conchiformis stesidae			
Elene laboriguap			
Elene laballae			
Elene laeagua			
Elene laorodens			
Neisseria shayegani			
Strogynocella sh			
Stenoxylacter acetivorans			
Vitreoscilla scitellensis			
Vitreoscilla stercoraia DSM513			
Crenobacter venae			
Crenobacter inestini			
Crenobacter luteus			
Crenobacter sedimenti			

Minor points.

1. Lines 152-155. This statement is not true for coccal Neisseria such as *N. gonorrhoeae*, *N. meningitidis*, *N. mucosa*, etc.
We agree and have corrected this.

2. Line 318 and elsewhere, make sure that *N. elongata* is correctly spelled.
We made sure that *N. elongata* is correctly spelled throughout the text

3. Line 320. It is not necessary to “validate” RNA-Seq results with real-time RT-PCR.
We agree and replaced “validated” with “confirmed”.

4. Line 376. This statement is too strong. Given the results presented, it is not necessarily true that the genes identified “contributed to the evolution of bacterial multicellularity”.
Reword to add a qualifier, such as “likely contributed” or “may have contributed”.
We agree and added “likely”.

5. Line 401. This sentence is awkward. Either delete “to” or change “resemble” to “similar”.
We deleted “to”.

Line 820. Indicate the source of fluorescent D-alanine analogs.
FDAAs were kindly provided by Michael vanNieuwenhze. This information has been included to the Methods section.

6. Fig. 1. Add a label for the blue cocci or define in the legend.
We corrected this.

7. Fig. 6b is not mentioned in the manuscript.
Fig.6b is now mentioned.

8. Fig. S7 needs more labels and explanation.
We agree and we have changed this figure accordingly (now Figure S10). Thank you!

Reviewer #3 (Remarks to the Author):

This manuscript provides landmark advances in our understanding the evolution of multicellular longitudinal cell division (MuLDi) in Neisseriaceae. By combining genome sequencing, microscopy and genetics they propose how this peculiar mode-of-division originated from an ancestral rod-shaped bacterium. Previous work from this lab pioneered

this fascinating mode of division, and here they build upon that previous work by focusing on oral cavity symbionts belonging to the Neisseriaceae. The microscopy in this paper is outstanding, and the finding that this mode-of-division originated from a rod-shaped ancestor is clearly demonstrated by the phylogenetic reconstruction. The authors towards the end of the paper pioneer the first steps in reconstructing this division mode in the rod-shaped Neisseria elongate, and by introducing key changes based on their thorough phylogenetic analysis they manage to increase the size of septa (which in MuLDi strains is larger). For their consideration, I have some suggestions that could improve this otherwise fantastic manuscript

1. I find the introduction, and in particular the first part (lines 56-86) quite intense in terms of the described organisms and their classification. There is a lot of information here on their morphology, classification, uni- versus multicellular etc (admittedly, I am not a phylogeny expert). Also, lines 83-86 quite abruptly start discussing fimbriae, which I understand later in the results, is important to determine the polarity. My advice would be to take out the fimbriae part in the introduction, and in the results section (lines 194-201) use one or two sentences that it is important to determine the polarity, for which antibodies against fimbriae are used.

We agree and removed the fimbriae part from the introduction.

2. Lines 169-174. I find this part oddly presented. I would start with the data (Fig. 1, 2, Fig. S1b-d, Sup Movies 1-4), and then compare to previous studies.

We edited the main text and hopefully improved its readability.

Also, some text could be added to describe these results without just pointing to the Figures/Movies

We agreed and revised accordingly.

3. In the parts describing how septa are formed in *A. filiformis*, *S. muelleri* and *C. steedae* I find myself switching between figures. Isn't it handier to organize the figures by species rather than imaging technique, in particular for the fluorescence panels? (lines 192-246)

We agree and Figures are now arranged by species for improved readability.

4. The parts on (the absence of) *mraZ* could be combined into one paragraph (so the RNAseq and the mutational analysis in *N. elongata*).

As we were not asked to reduce space, if the reviewer agree, we opted for maintaining inter-species RNAseq analysis with the remaining inter-species comparisons (gene presence/absence and AA permutations) and separate it from isogenic mutational analysis.

5. In the discussion the authors speculate on the transition from unicellular to multicellular longitudinal division. What I miss in the discussion is something related to the organization and segregation of DNA. This should have also evolved, or is this passively following morphological changes?

Thanks a lot for this question. A manuscript dealing with the configuration, conformation and segregation mode of *A. filiformis*, *S. muelleri* and *C. steedae* chromosomes is about to be submitted (Viehböck, Weber, Krause, Junier, Varoquaux, Boccad, Lioy and Bulgheresi. *Fixed chromosome configuration and multiple long-range chromosomal loops underlie genome organization in four animal symbionts*, in preparation). Given the formidably stable (*ori-ter*) configuration of the chromosomes of MuLDi bacteria (*ori* is proximal and *ter* is distal throughout the cell cycle irrespective of DNA segregation stage), our current hypothesis is that longitudinal division evolved to enable fixed chromosome configurations (i.e., virtually all genetic loci maintain their intracellular localization throughout the cell cycle).

And could this be something that (partly) explains why the authors have not managed to reconstruct this mode-of-division?

Yes, this could explain why we could not manipulate rods into MuLDi. Although we have not studied *N. elongata* chromosome configuration and segregation yet (or the chromosome of any

other transversally dividing rod-shaped *Neisseriaceae*), it is definitely on our things-to-do list, thank you!

Minor comments

Line 102: Ssg should be SsgB

Line 122: "...the acquisition OF amiC2..."

Line 139: "from THE NCBI database..."

We did all the suggested edits.

Line 154: light blue branches. To me all blue branches appear to have the same color.

We edited the Figure.

Line 272: ShIB. Add reference to this claim.

We added a reference (K. Poole, E. Schiebel and V. Braun J. Bacteriol., 170 (1988): 3177-3188).

Line 280. Why is *dgt*/*GloB* not discussed/mentioned?

It was an oversight not to mention them, we apologize. In the Results section of the revised manuscript, we have explained what the *dgt* and *gloB* genes encode for. *dgt* encodes for a dGTPase (Beauchamp, B. B., and C. C. Richardson.1988) and *gloB* for a hydroxyacylglutathione hydrolase that hydrolyzes S-D-lactoyl-glutathione into glutathione and D-lactic acid involved in methylglyoxal detoxification (O'Young, Sukdeo, Honek. Archives of Biochemistry and Biophysics. 2007). In *E. coli*, the loss of the former was shown to be mutagenic (Gawel et al., 2008) and that of the latter to impair resistance to the naturally occurring cytotoxic compound methylglyoxal (Reiger et al., 2015). Given that we currently cannot relate the loss of *dgt* and *gloB* with the evolution of multicellularity, we have decided not to Discuss them.

Line 291: perhaps best to say that there is a correlation in the mode of division and the presence/absence of certain genes

We rephrased accordingly.

Line 311-312. I think again this is correlation (so absence of *mraZ* correlates with different expression).

We rephrased accordingly.

Line 334: data not shown: I think it's common to include this in Suppl. Information nowadays.

We agree and added the data (see Figure S11 of revised manuscript). Thank you.

Reviewer #4 (Remarks to the Author):

In this article entitled "Evolution of multicellular longitudinally dividing oral cavity symbionts (*Neisseriaceae*)", Nyongesa et al. present a model of cell division of the multicellular longitudinally dividing (MuLDi) organisms in *Neisseriaceae* and propose an evolutionary scenario of the emergence of such phenotype. First, the authors present a phylogenomic analysis that describe the diversity in term of shape and way to divide in *Neisseriaceae* and conclude that MuLDi clades emerged likely from a rod-shape ancestor. Then, they characterize the structure of the peptidoglycan using microscopy and mass spectrometry. They also describe the dynamic of muropeptide insertion into the mid-cell and conclude that the separation of the daughter cells is performed by PG sheet insertion, asynchronously for *A. filiformis* and synchronously for *S. muelleri* and *C. steedae*. Next, they identified the key evolutionary events that occurred at the stems of both MuLDi clades,

involving notably major cell division and cell wall synthesis genes. Finally, the authors study deeper the effect of such evolutionary events by generating mutants of *mraZ* and other cell division genes that were gained, lost or mutated. The known diversity of prokaryotes increased dramatically in the last decades, although most of the cell processes (and cell division) are still studied in a very few model organisms (rod-shape for most of them). In this context, the study of cell division mechanisms in non-model and non-rod-shaped bacteria is of particular interest. The authors combined elegantly phylogenomics, cell biology and biochemistry to decipher both mechanism and evolution of the cell division of the MuLDi. The results are interesting and convincing and the text is globally well-written. Nevertheless, as a specialist of phylogenomics, I have a few recommendations to the authors to strengthen the genomic analysis and to better clarify some paragraphs.

Methodological

comments:

The methods for the genomic analysis are not very clear (l. 131-147 and 943-979), there are some discrepancies between supplementary tables and the text, and some information/data are missing. To clarify this:

- Write an entire section of M&M with the detail of the method of the reduction of the number of genomes from 21+365 genomes to 69 genomes (Download of NCBI genomes, ANI, clustering method, selection of the best genome per cluster of species).

We apologize for not being sufficiently clear and revised accordingly. The method of the reduction of the number of genomes to 75 genomes is now described in the results section of the manuscript.

- To make it easier for non-specialists, explain more clearly the rationale for each step, in both result and M&M sections (Why sequencing these 21 genomes, why mixing them with NCBI genomes, why comparing genomes with ANI, why reducing the number of genomes, ...)

This is now described in the Results section of the manuscript. The aim was to keep only the "best" genome for every species. In the NCBI database, there are several low-quality genomes (i.e., highly fragmented) and strains or species wrongly identified. We needed to be sure that we worked with a cured and clean database. Of note, regardless of the number of genome sequencing projects, there are only 5 species (=5 strains) of MuLDi in the *Neisseriaceae* family and they are all described here (Figure 1 and S4c and S6e).

- Indicate the software that has been used for ANI and the parameters in M&M. Also, give the parameters used for each software (Roary, MycoHIT, CapriB, ..).

Software and parameters have been provided in the revised manuscript (Methods and Tables).

- Provide the raw data of phylogenetic analysis (Alignments, Supermatrix, Tree with bootstrap values). Give the list of markers (core genome) in a table, the total number of markers and the number of positions of the alignment that have been used to build the phylogeny of *Neisseriaceae*. Also, provide the exact evolutionary model that has been selected by IQ-TREE.

The missing information is provided in Supplementary File 1 and in the revised Methods section and Tables.

- Indicate clearly the source (collection like DSMZ or samples?) of the newly sequenced genomes, in both result and M&M sections.

We have added the DMSZ or ATCC reference number in the Table S1, next to the species name.

- Highlight clearly the 21 genomes in the table S2.

We have highlighted the genome in Table S1 of the revised Manuscript.

- Table S1 and S2 seem to be inverted in the text (l. 137 and 140).

We corrected this.

- L. 109: I see only 41 genomes in table S2, not 42.

Thank you, we have corrected this

- I suppose that the 69-41 = 28 genomes correspond to the cocci clade 1. List these genomes in table S2 and indicate clearly that 69-41 corresponds to the number of genomes inside this clade in the text.

If the reviewer agrees, we would prefer not to include this information in this manuscript as some genomes are not yet public and some species are not described yet. As we excluded the cocci clade 1 from all the analyses, except the phylogenetic one (Figure 1), we feel that not including the cocci genomes does not impact the manuscript.

- Table S1: there are only 261 genomes instead of 365+21 genomes. Complete the matrix (even if ANI <70%).

We are sorry for this mistake. 261 genomes are the number of genomes that remains after excluding the many duplicates for *Neisseria meningitidis* and *Neisseria gonorrhoeae*. This is now stated in the manuscript.

Also, provide a list of the 365 genomes originating from NCBI with proper identifiers.

We have provided the strains in Table S2 of the revised manuscript.

The authors did not use a statistical method to infer the ancestral states (rod-shaped/cocci/MuLDi) (l. 158-161). The inference of the ancestral states using a Maximum Likelihood method such as PastML could reinforce the data. It is an easy and quick analysis and I have no doubt that the results of such method would be in agreement with the manual inferences presented in Figure 1. The only potential issue would be the 8 lineages for which there is no microscopy data, but they could be removed from the tree to perform the analysis.

Thanks a lot for this comment. The strains that we have not analyzed morphologically by ultrastructural analysis were already described as rod-shaped bacteria (except two species *N. sp. KEM232* and *N. sp. 83E034* for which we cannot acquire morphological information due to the unavailability of these strains). We have clarified this and added the appropriate references to the legend of Figure 1 of the revised manuscript. Furthermore, in agreement with the Reviewer, we have used PastML and confirmed that MuLDi and cocci evolved from a rod-shape ancestor.

Also, the authors could clearly indicate each genome for which microscopy data are available in table S2.

We agree and we have added this information to Table S2 of the revised Manuscript.

In any case, the argument of the wide distribution of rod-shaped species is not sufficient to infer the rod-shape nature of the ancestor. The rod-shape nature of basal lineages (outgroup, *E. corrodens* clade, *V. massiliensis* clade, ...) is crucial to such conclusion and this should be stated in the text.

We agree and have now used PastML to corroborate our conclusions.

Concerning the pangenome analysis (l. 958-959 in M&M), the use of the set of proteins of two reference genomes can be hazardous as some proteins could be only absent in the reference genomes but still widely present in the other genomes. The use of a simple method without a priori like whole database clustering methods (MMseqs2, SiLiX, HiFiX, Roary) would be more suitable for such analysis.

We understand your concerns. However, first, to avoid problems of annotation in the reference genome, we verified our analyses using reference genomes other than those of *Simonsiella muelleri* and *Neisseria elongata* (such as *Conchiformibius kuhniae* and *Kingella oralis*). This confirmed the events detected, but did not lead to the detection of more events. Apart from the reference genomes, it is important to note that we used an annotation-free TBlastN method (protein from the reference against translated genomes) to minimize bias due to annotation in the other genomes used. In this study, our aim was to search for proteins that are specifically present in

MuLDi and specifically absent in the rod-shaped *Neisseriaceae*. We agree that we could have arbitrarily lowered the cut-off (such as genes present in 90% of one group and absent in 90% of the other group), but this would have increased the number of false positives (or the background) so much to impair the identification of genetic events underlying the rod-to-MuLDi evolution.

We also tried to use alternative methods based on clustering methods such as Roary, Get_Homologues or clustalX (that are well-suited to find core proteomes to be used for phylogeny and inter-species RNAseq analyses) to verify gene presence and absence. However, the results obtained with such methods did not pinpoint lineage-specific genes. Indeed, when verifications were made using tblastn or gene synteny of individual loci, a great proportion of these results were false positives.

In conclusion, also considering successful previous applications (the original study of HGT in *Mycobacteria* and the loss of ZapD in C1), MycoHIT (with stringent cut-off, to avoid false positives or background) appears to be a valid and suitable tool for the detection of genetic events underlying evolutionary transitions. Nevertheless, we acknowledge (both here and in the Discussion) that this method cannot determine all genetic events.

Other comments:

I. 152-155: This sentence and especially “Neisseriaceae are rod-shaped, except for two closely related species” is misleading as the figure and the text I. 163-165 highlight not only this clade as an exception, but two cocci clades and two MuLDi clades. Please rephrase.
We rephrased, thank you.

I. 155-156: Please explain very shortly the method that has been used to infer the loss of such genes in the text.
We have briefly explained the method (lines XXX).

Figure 1: I have a few comments:

- **The difference between light and dark blue is not visible.**
- **One microscopy image is not connected to any tip.**
- **Use 1 and 2 for both cocci and MuLDi clades is confusing, maybe 1/2 and A/B?**

We improved Figure 1 but we now refer to the two MuLDi *Neisseriaceae* lineages with M1 and M2 and we refer to the two coccoid *Neisseriaceae* lineages with C1 and C2.

I. 180-181: This model of division is really intriguing and rises some questions. Does it mean that cells are encapsulated by the outer membrane like a “plastic film of a six-pack of bottles”, or is it possible that the outer membrane is still present in the mid-cell?

Thanks a lot for this comment. Cells belonging to the same filament share the outer membrane (OM), likely due to a delayed OM invagination coupled with thickening/splitting of the septal peptidoglycan (see response to Reviewer #1 and TEM image in this Rebuttal). We have also added higher magnification TE micrographs (see revised Figure 2), clarified this in the main text and added a schematic representation of the OM, PG and IM (see Figure S2 of revised manuscript).

The first hypothesis would mean that another secondary division occurs (when the cells are really separated during the chain growth) to lysate the peptidoglycan and invaginate the outer membrane. In other model bacteria, the invagination of the outer membrane is driven by septation. So, what could be the driving force for such a case? I think that such hypotheses/questions should be developed in the discussion section.

Our ultrastructural analysis suggests that delayed OM invagination occurs in addition to a PG split into two sheets. During this secondary step the PG appears to double its thickness (see TEM image accompanying response to Reviewer #1, this Rebuttal). Therefore, in agreement with you, we believe that a secondary round of “division” drives the invagination of the outer membrane. We have added this hypothesis to the discussion.

I. 210: “with what “is” observed”?

By 3D-SIM, we have clarified this.

I. 264 and 469: Choose either phosphatidate cytidylyltransferase or CDP-diacylglycerol synthase to describe CdsA function.

We chose phosphatidate cytidylyltransferase.

I. 445: For your information, it has been suggested that the presence/absence of MreB and Min genes is responsible for the rod-shape/cocci phenotype, and not particularly the fragmentation of the DCW cluster (How to Build a Bacterial Cell: MreB as the Foreman of E. coli Construction, Shi et al. 2018, A Comprehensive Evolutionary Scenario of Cell Division and Associated Processes in the Firmicutes, Garcia et al., 2020).

We agree and did cite Shi et al. 2018.

I. 478: Another key event that could have occurred is the rearrangement of the order of genes within the genome. It could be interesting to explore this point (But I do not ask the authors to do it).

We agree with the reviewer and we hope to test the effect of gene order rearrangement in the very near future. Thank you.

Reviewers' Comments:

Reviewer #1:

Remarks to the Author:

I am very satisfied with the authors' responses to my suggestions. My only additional suggestion is to replace "inner membrane" or "IM" with "cytoplasmic membrane" or "CM" because my interpretation of the results is that the "IM" is a canonical cytoplasmic membrane. Great paper!

Brian Hedlund

Reviewer #2:

Remarks to the Author:

The authors have done an excellent job of revising the manuscript, and all of my concerns have been adequately addressed.

Reviewer #4:

Remarks to the Author:

The authors addressed all my concerns, I have no additional remark.

Point-to-point rebuttal

Reviewer #1 (Remarks to the Author):

I am very satisfied with the authors' responses to my suggestions. My only additional suggestion is to replace "inner membrane" or "IM" with "cytoplasmic membrane" or "CM" because my interpretation of the results is that the "IM" is a canonical cytoplasmic membrane. Great paper! Brian Hedlund

We agree with the Reviewer and replaced inner membrane (IM) with cytoplasmic membrane (CM) throughout the manuscript.

Reviewer #2 (Remarks to the Author):

The authors have done an excellent job of revising the manuscript, and all of my concerns have been adequately addressed.

Thank you.

Reviewer #4 (Remarks to the Author):

The authors addressed all my concerns, I have no additional remark.

Thank you.